# GAUCHE: A Library for Gaussian Processes in Chemistry

**Ryan-Rhys Griffiths**[1*]    **Leo Klarner**[2*]    **Henry Moss**[3*]    **Aditya Ravuri**[3*]    **Sang Truong**[4*]
**Samuel Stanton**[5*]    **Gary Tom**[6,7*]    **Bojana Rankovic**[8,9*]    **Yuanqi Du**[10*]    **Arian Jamasb**[3*]

**Aryan Deshwal**[11]    **Julius Schwartz**[3]    **Austin Tripp**[3]    **Gregory Kell**[12]    **Simon Frieder**[2]
**Anthony Bourached**[13]    **Alex J. Chan**[3]    **Jacob Moss**[3]    **Chengzhi Guo**[3]
**Johannes Durholt**[14]    **Saudamini Chaurasia**[15]    **Ji Won Park**[5]    **Felix Strieth-Kalthoff**[6]

**Alpha A. Lee**[3]    **Bingqing Cheng**[16]    **Alán Aspuru-Guzik**[6,7,17]    **Philippe Schwaller**[8,9]
**Jian Tang**[18,19,17]

[1]Meta    [2]University of Oxford    [3]University of Cambridge    [4]Stanford University    [5]Genentech
[6]University of Toronto    [7]Vector Institute    [8]EPFL    [9]NCCR Catalysis    [10]Cornell University
[11]Washington State University    [12]King's College London    [13]University College London
[14]Evonik Industries AG    [15]Syracuse University    [16]IST Austria    [17]CIFAR AI Research Chair
[18]MILA Quebec AI Institute    [19]HEC Montreal

[*] Equal contributions

{ryangriff123,leojklarner}@gmail.com

## Abstract

We introduce GAUCHE, an open-source library for GAUssian processes in CHEmistry. Gaussian processes have long been a cornerstone of probabilistic machine learning, affording particular advantages for uncertainty quantification and Bayesian optimisation. Extending Gaussian processes to molecular representations, however, necessitates kernels defined over structured inputs such as graphs, strings and bit vectors. By providing such kernels in a modular, robust and easy-to-use framework, we seek to enable expert chemists and materials scientists to make use of state-of-the-art black-box optimization techniques. Motivated by scenarios frequently encountered in practice, we showcase applications for GAUCHE in molecular discovery, chemical reaction optimisation and protein design.

The codebase is made available at https://github.com/leojklarner/gauche.

## 1   Introduction

Early-stage scientific discovery is often characterised by the limited availability of high-quality experimental data [1, 2, 3], meaning that there is much knowledge to gain from targeted experiments. As such, machine learning methods that facilitate discovery in low data regimes, such as Bayesian optimisation (BO) [4, 5, 6, 7, 8, 9] and active learning (AL) [10, 11], have great potential to expedite the rate at which useful molecules, materials, chemical reactions and proteins can be discovered.

At present, Bayesian neural networks (BNNs) and deep ensembles are typically the method of choice to generate uncertainty estimates for molecular BO and AL loops [10, 12, 13, 14]. For small datasets, however, Gaussian processes (GPs) may often be a preferable and more appropriate choice [15, 16]. Furthermore, GPs possess particularly advantageous properties for BO; first, they admit exact as opposed to approximate Bayesian inference and second, few of their parameters need to be determined by hand. In the words of Sir David MacKay [17],

> "Gaussian processes are useful tools for automated tasks where fine tuning for each problem is not possible. We do not appear to sacrifice any performance for this simplicity."

37th Conference on Neural Information Processing Systems (NeurIPS 2023).

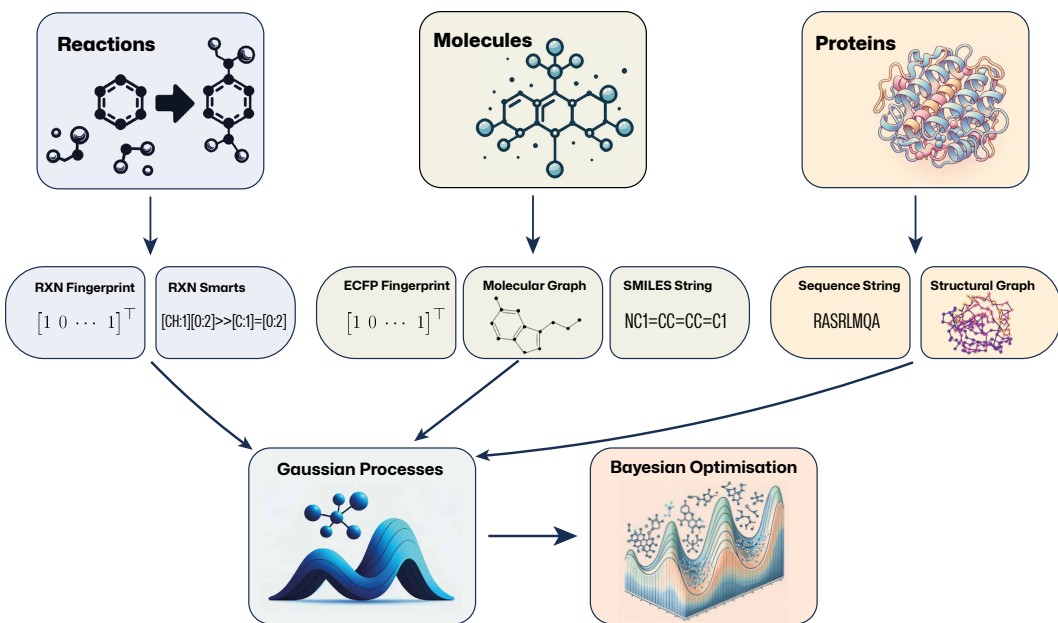

Figure 1: An overview of the applications and representations available in GAUCHE.

The iterative model refitting required in BO makes it a prime example of such an automated task. However, canonical GPs typically assume continuous input spaces of low and fixed dimensionality, hindering their application to standard molecular representations such as SMILES/SELFIES strings [18, 19, 20], topological fingerprints [21, 22, 23] and discrete graphs [24, 25].

With GAUCHE, we provide a modular, robust and easy-to-use framework to rapidly prototype GPs with 30+ GPU-accelerated string, fingerprint and graph kernels that operate on a range of molecular representations (see Figure 1). Furthermore, GAUCHE interfaces with the GPyTorch [26] and BoTorch [27] libraries and contains an extensive set of tutorial notebooks to make state-of-the-art probabilistic modelling and black-box optimization techniques more easily accessible to scientific experts in chemistry, materials science and beyond.

## 2 Background

We briefly recall the fundamentals of Gaussian processes and Bayesian optimisation in Sections 2.1 and 2.2, respectively, and refer the reader to [28] and [29, 30, 31] for a more comprehensive treatment.

### 2.1 Gaussian Processes

**Notation**  $\mathbf{X} \in \mathbb{R}^{n \times d}$ is a design matrix of $n$ training examples of dimension $d$. A given row $i$ of the design matrix contains a training molecule's representation $\mathbf{x}_i$. A GP is specified by a mean function, $m(\mathbf{x}) = \mathbb{E}[f(\mathbf{x})]$ and a covariance function $k(\mathbf{x}, \mathbf{x}') = \mathbb{E}[(f(\mathbf{x}) - m(\mathbf{x}))(f(\mathbf{x}') - m(\mathbf{x}'))]$. $K_\theta(\mathbf{X}, \mathbf{X})$ is a kernel matrix, where entries are computed by the kernel function as $[K]_{ij} = k(\mathbf{x}_i, \mathbf{x}_j)$ and $\theta$ represents the set of kernel hyperparameters. The GP specifies the full distribution over the function $f$ to be modelled as

$$f(\mathbf{x}) \sim \mathcal{GP}\big(m(\mathbf{x}), k(\mathbf{x}, \mathbf{x}')\big).$$

**Training**  Hyperparameters for GPs comprise kernel hyperparameters, $\theta$, in addition to the likelihood noise, $\sigma_y^2$. These hyperparameters are chosen by optimising an objective function known as the negative log marginal likelihood (NLML)

$$\log p(\mathbf{y}|\mathbf{X}, \theta) = \underbrace{-\frac{1}{2}\mathbf{y}^\top (K_\theta(\mathbf{X}, \mathbf{X}) + \sigma_y^2 I)^{-1}\mathbf{y}}_{\text{encourages fit with data}} \underbrace{-\frac{1}{2}\log|K_\theta(\mathbf{X}, \mathbf{X}) + \sigma_y^2 I|}_{\text{controls model capacity}} -\frac{N}{2}\log(2\pi),$$

where $\sigma_y^2 I$ represents the variance of i.i.d. Gaussian noise on the observations $\mathbf{y}$. The NLML embodies Occam's razor for Bayesian model selection [28] in favouring models that fit the data without being overly complex.

**Prediction** At test locations $\mathbf{X}_*$, assuming a zero mean function obtained following the standardization of the outputs $\mathbf{y}$, the GP returns a predictive mean, $\bar{\mathbf{f}}_* = K(\mathbf{X}_*, \mathbf{X})[K(\mathbf{X}, \mathbf{X}) + \sigma_y^2 I]^{-1} \mathbf{y}$, and a predictive uncertainty $\text{cov}(\mathbf{f}_*) = K(\mathbf{X}_*, \mathbf{X}_*) - K(\mathbf{X}_*, \mathbf{X})[K(\mathbf{X}, \mathbf{X}) + \sigma_y^2 I]^{-1} K(\mathbf{X}, \mathbf{X}_*)$.

## 2.2 Bayesian Optimisation

In molecular discovery campaigns, we are typically interested in solving problems of the form

$$\mathbf{x}^\star = \arg \max_{\mathbf{x} \in \mathcal{X}} f(\mathbf{x}),$$

where $f(\cdot) : \mathcal{X} \to \mathbb{R}$ is an expensive black-box function over a structured input domain $\mathcal{X}$. In our setting the structured input domain consists of a set of molecular representations (graphs, strings, bit vectors) and the expensive black-box function is an experimentally determined property of interest that we wish to optimise. Bayesian optimisation (BO) [32, 33, 34, 35, 29, 36] is a data-efficient methodology for determining $\mathbf{x}^\star$. BO operates sequentially by selecting input locations at which to query the black-box function $f$ with the aim of identifying the optimum in as few queries as possible. Evaluations are focused on promising areas of the input space as well as areas with high uncertainty—a balancing act known as the exploration/exploitation trade-off.

The two components of a BO scheme are a probabilistic surrogate model and an acquisition function. The surrogate model is typically chosen to be a GP due to its ability to maintain calibrated uncertainty estimates through exact Bayesian inference. The uncertainty estimates of the surrogate model are then leveraged by the acquisition function to propose new input locations to query. The acquisition function is a heuristic that trades off exploration and exploitation, well-known examples of which include expected improvement (EI) [33, 35] and entropy search [37, 38, 39, 40]. After the acquisition function proposes an input location, the black-box is evaluated at that location, the surrogate model is retrained and the process is repeated until a solution is obtained.

## 3 Molecular Representations

We review commonly used representations for molecules (Section 3.1), chemical reactions (Section 3.2) and proteins (Section 3.3), before describing the kernels that operate on them in Section 4. An overview of the representations considered by GAUCHE is provided in Figure 1.

### 3.1 Molecules

**Graphs** A molecule can be represented as an undirected, labelled graph $\mathcal{G} = (\mathcal{V}, \mathcal{E})$ where vertices $\mathcal{V} = \{v_1, \ldots, v_N\}$ represent the atoms of an $N$-atom molecule and edges $\mathcal{E} \subseteq \mathcal{V} \times \mathcal{V}$ represent covalent bonds between these atoms. Additional information may be incorporated in the form of vertex and edge labels $\mathcal{L} : \mathcal{V} \times \mathcal{E} \to \Sigma_V \times \Sigma_E$ by specifying e.g. atom types or bond orders.

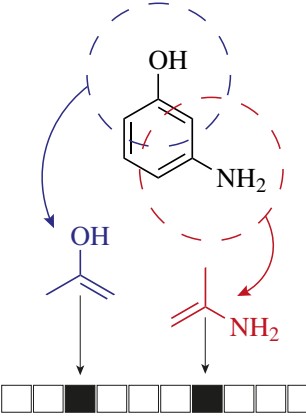

**Fingerprints** Molecular fingerprints enumerate sets or bags of subgraphs $\mathcal{G}' = (\mathcal{V}' \subseteq \mathcal{V}, \mathcal{E}' \subseteq \mathcal{E})$ of a certain type and then hash them into machine-readable bit or count vectors. Extended connectivity fingerprints (ECFPs) [21], for example, enumerate all circular subgraphs up to a pre-specified radius parameter by assigning initial numeric identifiers to each atom in a molecule and iteratively updating them based on the identifiers of their neighbours. Each level of iteration appends substructural features of increasing non-locality to an array, which is then hashed to a bit vector reflecting the presence or absence of those substructures in the molecule (see Figure 2). We choose a radius of 3 for all experiments in the main text and provide a more detailed ablation of the radius parameter in Appendix E.3.

Figure 2: Visualisation of the ECFP subgraph enumeration and hashing procedures.

Additionally, we make use of fragment descriptors, which are count vectors in which each component indicates the count of a particular functional group present in a molecule, as well as the concatenation of the fingerprint and fragment feature vectors, a representation termed fragprints [3], which has shown strong empirical performance. Example representations of $\mathbf{x_{fp}}$ for fingerprints, $\mathbf{x_{fr}}$ for fragments and $\mathbf{x_{frp}}$ for fragprings are given as

$$\mathbf{x_{fp}} = \begin{bmatrix} 1 & 0 & \cdots & 1 \end{bmatrix}^\top \quad \mathbf{x_{fr}} = \begin{bmatrix} 3 & 0 & \cdots & 2 \end{bmatrix}^\top \quad \mathbf{x_{frp}} = \begin{bmatrix} 1 & 0 & \cdots & 1 & 3 & 0 & \cdots & 2 \end{bmatrix}^\top$$

**Strings** The Simplified Molecular-Input Line-Entry System (SMILES) is a text-based representation of molecules [18, 19], examples of which are given in Figure 3. Self-Referencing Embedded Strings (SELFIES) [20] is an alternative string representation to SMILES such that a bijective mapping exists between a SELFIES string and a molecule.

Figure 3: SMILES strings for structurally similar molecules. Similarity is encoded in the string through common contiguous subsequences (black). Local differences are highlighted in red.

## 3.2 Reaction Representations

Chemical reactions consist of (multiple) reactants and reagents that react to form one or more products. The choice of reactant/reagent typically constitutes a categorical design space. Taking as an example the high-throughput experiments by [41] on Buchwald-Hartwig reactions, the reaction design space consists of 15 aryl and heteroaryl halides, 4 Buchwald ligands, 3 bases, and 23 isoxazole additives.

**Concatenated Molecular Representations** If the number of reactants and reagents is constant, the molecular representations discussed in Section 3.1 may be used to represent them, and the vectors for the individual reaction components can be concatenated to construct a representation of the reaction [41, 42]. An additional and commonly used concatenated representation is the one-hot-encoding (OHE) of the reaction categories where bits specify which of the components in the different reactant and reagent categories is present. In the Buchwald-Hartwig example, the OHE would describe which of the aryl halides, Buchwald ligands, bases and additives are used in the reaction, resulting in a 44-dimensional bit vector [43].

**Differential Reaction Fingerprints** Inspired by the hand-engineered difference reaction fingerprints by [44] and [45] recently introduced the differential reaction fingerprint (DRFP). This reaction fingerprint is constructed by taking the symmetric difference of the sets containing the molecular substructures on both sides of the reaction arrow. Reagents are added to the reactants. The size of the reaction bit vector generated by DRFP is independent of the number of reaction components.

**Data-Driven Reaction Fingerprints** [46] described data-driven reaction fingerprints using Transformer models such as BERT [47] trained in a supervised or an unsupervised fashion on reaction SMILES. Those models can be fine-tuned on the task of interest to learn more specific reaction representations [48] (RXNFP). Similar to the DRFP, the size of the data-driven reaction fingerprints is independent of the number of reaction components.

## 3.3 Protein Representations

Proteins are large macromolecules that adopt complex 3D structures. They can be represented in string form describing the underlying amino acid sequence. Graphs at varying degrees of coarseness may also be used for structural representations that capture spatial and intramolecular relationships between structural elements, such as atoms, residues, secondary structures and chains. GAUCHE interfaces with Graphein [49], a library for pre-processing and computing graph representations of structural biological data, thereby enabling the application of graph kernel-based methods to protein structure. We provide experiments on protein fitness prediction in Appendix E.1.

# 4 Molecular Kernels

The choice of kernel is an important inductive bias for the properties of the function being modelled. A common choice for continuous input domains is the radial basis function kernel

$$k_{\text{RBF}}(\mathbf{x}, \mathbf{x}') = \sigma_f^2 \exp\left(\frac{-||\mathbf{x} - \mathbf{x}'||_2^2}{2\ell^2}\right),$$

where $\sigma_f^2$ is the signal amplitude hyperparameter (vertical lengthscale) and $\ell$ is the (horizontal) lengthscale hyperparameter. However, in order to train GPs on the molecular representations covered in Section 3, bespoke kernel functions that are able to operate non-continuous input spaces are needed.

## 4.1 Fingerprint Kernels

**Scalar Product Kernel**   The simplest kernel to operate on fingerprints is the scalar product or linear kernel defined for vectors $\mathbf{x}, \mathbf{x}' \in \mathbb{R}^d$ as

$$k_{\text{Scalar Product}}(\mathbf{x}, \mathbf{x}') \coloneqq \sigma_f^2 \cdot \langle \mathbf{x}, \mathbf{x}' \rangle,$$

where $\sigma_f$ is a scalar signal variance hyperparameter and $\langle \cdot, \cdot \rangle$ is the Euclidean inner product.

**Tanimoto Kernel**   Introduced as a general similarity metric for binary attributes [50], the Tanimoto kernel was first used in chemoinformatics in conjunction with non-GP-based kernel methods [51]. It is defined for binary vectors $\mathbf{x}, \mathbf{x}' \in \{0, 1\}^d$ for $d \geq 1$ as

$$k_{\text{Tanimoto}}(\mathbf{x}, \mathbf{x}') \coloneqq \sigma_f^2 \cdot \frac{\langle \mathbf{x}, \mathbf{x}' \rangle}{\|\mathbf{x}\|^2 + \|\mathbf{x}'\|^2 - \langle \mathbf{x}, \mathbf{x}' \rangle},$$

where $|| \cdot ||$ is the Euclidean norm.

In addition to the Tanimoto kernel, GAUCHE provides parallelisable and batch-GP-compatible implementations of 12 other bit and count vector kernels that are presented in Appendix G.

## 4.2 String Kernels

String kernels [52, 53] measure the similarity between strings by examining the degree to which their sub-strings differ. In GAUCHE, we implement the SMILES string kernel [54] which calculates an inner product between the occurrences of sub-strings, considering all contiguous sub-strings made from at most $n$ characters (we set $n = 5$ in our experiments). Therefore, for the sub-string count featurisation $\phi : \mathcal{S} \to \mathbb{R}^p$, also known as a bag-of-characters representation [55], the SMILES string kernel between two strings $\mathcal{S}$ and $\mathcal{S}'$ is given by

$$k_{\text{String}}(\mathcal{S}, \mathcal{S}') \coloneqq \sigma_f^2 \cdot \langle \phi(\mathcal{S}), \phi(\mathcal{S}') \rangle.$$

More complicated string kernels do exist in the literature, for example, GAUCHE also provides an implementation of the subset string kernel [56] which allows non-contiguous matches. However, we found that the significant added computational cost of these methods did not provide improved performance over the more simple SMILES string kernel in the context of molecular data. Note that although named the SMILES string kernel, this kernel can also be applied to any other string representation of molecules e.g. SELFIES or protein sequences.

## 4.3 Graph Kernels

Graph kernels define a mapping $\phi_\lambda : \mathcal{G} \to \mathcal{H}$ from a graph domain $\mathcal{G}$ to a feature space $\mathcal{H}$, in which the inner product between a pair of graphs $g, g' \in \mathcal{G}$ serves as a similarity measure

$$k_{\text{Graph}}(g, g') \coloneqq \sigma_f^2 \cdot \langle \phi_\lambda(g), \phi_\lambda(g') \rangle_{\mathcal{H}},$$

where $\lambda$ denotes kernel-specific hyperparameters. Depending on how $\phi_\lambda$ is defined, the kernel captures different substructural motifs and is characterised by different hyperparameters. The Weisfeiler-Lehman (WL) kernel [57], for instance, is given by the inner products of label count vectors over $\lambda$ iterations of the Weisfeiler-Lehman algorithm [58].

To maximise the number of graph kernels available in GAUCHE we implemented the SIGP class, which enables PyTorch-based GPs to be trained on non-tensorial inputs with any kernel from the GraKel library [59] (see Appendix F for more details).

Table 1: Predictive accuracy and calibration of Gaussian process and probabilistic deep learning models across four different molecular property prediction benchmarks. RMSE ($\downarrow$) and NLPD ($\downarrow$) are reported as mean and standard error over 20 random 80/20 train/test splits. The best models up to statistical significance are highlighted in bold. Numerical issues were encountered with the WL kernel on the large lipophilicity dataset and the corresponding entries are left blank.

| | Model | Representation | Photoswitch | | ESOL | | FreeSolv | | Lipophilicity | |
|---|---|---|---|---|---|---|---|---|---|---|
| | | | RMSE | NLPD | RMSE | NLPD | RMSE | NLPD | RMSE | NLPD |
| **Gaussian Processes** | Tanimoto | fragprints | $\mathbf{20.9_{\pm0.7}}$ | $\mathbf{0.22_{\pm0.03}}$ | $0.71_{\pm0.01}$ | $0.33_{\pm0.01}$ | $1.31_{\pm0.06}$ | $0.28_{\pm0.02}$ | $\mathbf{0.67_{\pm0.01}}$ | $\mathbf{0.71_{\pm0.01}}$ |
| | | fingerprints | $23.4_{\pm0.8}$ | $0.33_{\pm0.03}$ | $1.01_{\pm0.01}$ | $0.71_{\pm0.01}$ | $1.93_{\pm0.09}$ | $0.58_{\pm0.03}$ | $0.76_{\pm0.01}$ | $0.85_{\pm0.01}$ |
| | | fragments | $26.3_{\pm0.8}$ | $0.50_{\pm0.04}$ | $0.91_{\pm0.01}$ | $0.57_{\pm0.01}$ | $1.49_{\pm0.05}$ | $0.44_{\pm0.03}$ | $0.80_{\pm0.01}$ | $0.94_{\pm0.02}$ |
| | Scalar Product | fragprints | $22.5_{\pm0.7}$ | $\mathbf{0.23_{\pm0.03}}$ | $0.88_{\pm0.01}$ | $0.53_{\pm0.01}$ | $1.27_{\pm0.02}$ | $0.25_{\pm0.02}$ | $0.77_{\pm0.01}$ | $0.92_{\pm0.01}$ |
| | | fingerprints | $24.8_{\pm0.8}$ | $0.33_{\pm0.03}$ | $1.17_{\pm0.01}$ | $0.84_{\pm0.01}$ | $1.93_{\pm0.07}$ | $0.64_{\pm0.03}$ | $0.84_{\pm0.01}$ | $1.03_{\pm0.01}$ |
| | | fragments | $36.6_{\pm1.0}$ | $0.80_{\pm0.03}$ | $1.15_{\pm0.01}$ | $0.82_{\pm0.01}$ | $1.63_{\pm0.03}$ | $0.54_{\pm0.02}$ | $0.97_{\pm0.01}$ | $0.88_{\pm0.10}$ |
| | String | SMILES | $24.8_{\pm0.7}$ | $0.30_{\pm0.04}$ | $\mathbf{0.66_{\pm0.01}}$ | $\mathbf{0.29_{\pm0.03}}$ | $1.31_{\pm0.01}$ | $\mathbf{0.16_{\pm0.02}}$ | $\mathbf{0.68_{\pm0.01}}$ | $\mathbf{0.72_{\pm0.01}}$ |
| | WL Kernel | graph | $22.4_{\pm1.4}$ | $0.39_{\pm0.11}$ | $1.04_{\pm0.02}$ | $0.76_{\pm0.001}$ | $1.47_{\pm0.06}$ | $0.47_{\pm0.02}$ | - | - |
| **Neural Nets** | FC-BNN | fragprints | $\mathbf{20.9_{\pm0.6}}$ | $1.63_{\pm0.44}$ | $0.88_{\pm0.01}$ | $1.70_{\pm0.11}$ | $1.39_{\pm0.03}$ | $1.41_{\pm0.38}$ | $0.75_{\pm0.01}$ | $3.82_{\pm0.12}$ |
| | | fingerprints | $22.4_{\pm0.7}$ | $2.22_{\pm0.56}$ | $1.08_{\pm0.02}$ | $2.59_{\pm0.40}$ | $1.93_{\pm0.07}$ | $2.65_{\pm0.72}$ | $0.81_{\pm0.01}$ | $3.74_{\pm0.10}$ |
| | | fragments | $25.8_{\pm0.7}$ | $0.69_{\pm0.09}$ | $1.03_{\pm0.01}$ | $1.93_{\pm0.28}$ | $1.48_{\pm0.02}$ | $0.89_{\pm0.12}$ | $0.87_{\pm0.01}$ | $5.52_{\pm0.23}$ |
| | GNN-BNN | graph | $28.5_{\pm1.2}$ | $1.00_{\pm0.13}$ | $0.88_{\pm0.01}$ | $1.70_{\pm0.11}$ | $\mathbf{0.96_{\pm0.01}}$ | $1.01_{\pm0.02}$ | $0.73_{\pm0.02}$ | $1.14_{\pm0.01}$ |
| | CNN Ensemble | SELFIES | $26.4_{\pm1.0}$ | $4.34_{\pm0.55}$ | $\mathbf{0.67_{\pm0.01}}$ | $2.91_{\pm0.14}$ | $1.29_{\pm0.04}$ | $2.24_{\pm0.21}$ | $0.75_{\pm0.01}$ | $2.60_{\pm0.06}$ |
| | CNN DKL GP | SELFIES | $25.1_{\pm0.8}$ | $0.48_{\pm0.05}$ | $0.94_{\pm0.04}$ | $0.90_{\pm0.15}$ | $1.41_{\pm0.11}$ | $0.33_{\pm0.04}$ | $0.91_{\pm0.01}$ | $1.46_{\pm0.03}$ |

## 5 Experiments

We evaluate GAUCHE on a range of regression, uncertainty quantification (UQ) and Bayesian optimisation (BO) tasks. The principle goal in conducting regression and UQ benchmarks is to gauge whether performance on these tasks may be used as a proxy for BO performance. BO is a powerful tool for automated scientific discovery but one would prefer to avoid model misspecification in the surrogate when deploying a scheme in the real world. We make use of the following datasets with experimentally determined labels:

- **Photoswitch** The labels $y$ are the values of the $E$ isomer $\pi - \pi^*$ transition wavelength for 392 photoswitch molecules [3].
- **ESOL** The labels $y$ are the logarithmic aqueous solubility values for 1128 organic small molecules [60].
- **FreeSolv** The labels $y$ are the hydration free energies for 642 molecules [41].
- **Lipophilicity** The labels $y$ are the octanol/water distribution coefficient (log D at pH 7.4) of 4200 compounds curated from the ChEMBL database [61, 62].
- **Buchwald-Hartwig reactions** The labels $y$ are the yields for 3955 Pd-catalysed Buchwald–Hartwig C–N cross-couplings [41].
- **Suzuki-Miyaura reactions** The labels $y$ are the yields for 5760 Pd-catalysed Suzuki-Miyaura C-C cross-couplings [63].

### 5.1 Regression and Uncertainty Quantification (UQ)

**Experimental setup** For the regression and uncertainty quantification experiments, all datasets were randomly split into training and test sets with a ratio of 80/20. (Note that validation sets are not required for the GP models, since hyperparameters are chosen using the marginal likelihood objective on the train set). All GP models were trained using the L-BFGS-B optimiser [64] and, if not stated otherwise, the default settings in the GPyTorch and BoTorch libraries apply.

To quantify model performance, the predictive accuracy and the calibration of the predictive uncertainty estimates of the fitted models were evaluated on the held-out test set and summarised as the root-mean-square error (RMSE) and the negative log predictive density (NLPD), respectively. The mean and standard error of these metrics over 20 different random splits are reported in Table 1.

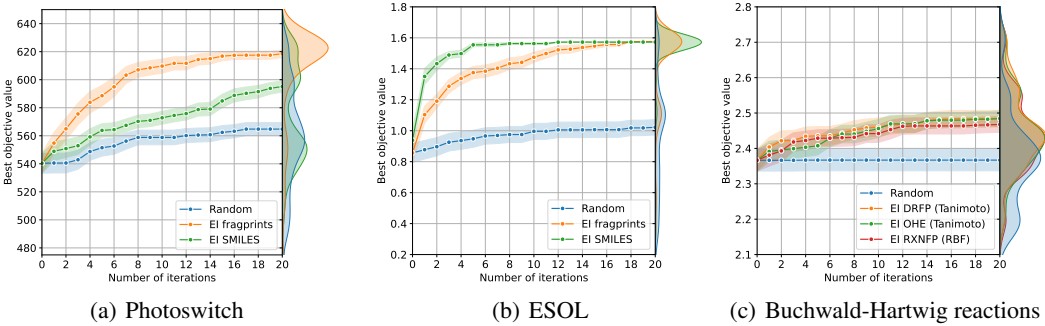

|  | (a) Photoswitch | (b) ESOL | (c) Buchwald-Hartwig reactions |

Figure 4: BO performance reporting the standard error from 50 randomly initialised trials (20 for Buchwald-Hartwig). A kernel density estimate over the trials is shown on the right axis. EI fragprints results use the Tanimoto kernel. The random search baseline is indicated in blue.

Alternative metrics to measure the quality of the predictive uncertainty estimates such as the mean standardised log loss (MSLL) and the quantile coverage error (QCE) are reported in Appendix A, while additional results for the reaction yield prediction datasets and scaffold splits are presented in Appendices B and C. We also benchmark the performance of GP models against a range of Bayesian neural network- (BNN) and deep ensemble-based methods that are detailed in Appendix D.

**Results** Summarising our results in Table 1, we find that Tanimoto-based GPs generally outperform Scalar Product ones in terms of RMSD and NLDP, while string kernel-based GPs often yield even better performance. We additionally note that the quality of the predictive uncertainty estimates roughly correlates with predictive accuracy in the case of GP-based models.

While deep probabilistic models attained competitive results in terms of RMSD, we found their uncertainty estimates to be consistently less reliable than those of GP-based models with discrete string kernels or shallow continuous kernels on hand-crafted features (e.g. fragprints), limiting their suitability for Bayesian optimisation and active learning. Our results suggest that for small to mid-sized molecular datasets the Tanimoto kernel combined with fragprint representations in particular is a very compelling option, with good accuracy, calibration, and runtime across all tasks.

### 5.2 Bayesian Optimisation

Building on these results, we employed the two best-performing kernels, namely the Tanimoto-fragprint kernel and the SMILES string kernel, to undertake Bayesian optimization BO over the photoswitch and ESOL datasets. BO is run for 20 iterations of sequential candidate selection (EI acquisition) where candidates are drawn from 95% of the dataset. The models are initialised with 5% of the dataset. In the case of the photoswitch dataset, this corresponds to just 19 molecules. The results are provided in Figure 4. In this ultra-low data setting—common to many areas of synthetic chemistry [3]—both models significantly outperform the random search baseline, highlighting the real-world use-case for such models in supporting human chemists to prioritise candidates for synthesis.

Furthermore, one may observe that BO performance is tightly coupled to regression and UQ performance. In the case of the photoswitch dataset, the better-performing Tanimoto model on regression and UQ also achieves better BO performance, while on the ESOL dataset, the string kernel performs best. Additionally, we run BO on the Buchwald-Hartwig dataset using the Tanimoto kernel for the bit-vector representations DRFP and OHE, and the RBF kernel for RXNFP. All three representations perform similarly and outperform the random search.

Finally, we also investigate the performance of GP-based models for preferential BO—a setting in which the acquisition strategy only requires rank-based preferences of candidates, as opposed to their absolute objective function values [65, 66, 67, 68]. We use a Tanimoto-fragprint kernel GP model to perform molecular Bayesian optimization on the photoswitch dataset using binary preference data alone and present the full results in Appendix E.4.

Table 2: An overview of existing open-source GP, BO and molecular machine learning libraries. With GAUCHE, we provide a modular, robust and easy-to-use framework that combines state-of-the-art probabilistic modelling and black-box optimization techniques with bespoke molecular representations and kernels to make them more easily accessible to the broader scientific community.

| Library | Gaussian Processes | Bayesian Optimisation | Molecular Representations | Chemistry Tutorials | Graph Kernels | Bit Vector Kernels | String Kernels |
|---|---|---|---|---|---|---|---|
| GPyTorch [26] | ✓ | ✗ | ✗ | ✗ | ✗ | ✗ | ✗ |
| GPflow [69, 70] | ✓ | ✗ | ✗ | ✗ | ✗ | ✗ | ✗ |
| BoTorch [27] | ✓ | ✓ | ✗ | ✗ | ✗ | ✗ | ✗ |
| DeepChem [71] | ✗ | ✗ | ✓ | ✓ | ✗ | ✗ | ✗ |
| GraKel [59] | ✗ | ✗ | ✗ | ✗ | ✓ | ✗ | ✗ |
| FlowMO [72] | ✓ | ✗ | ✓ | ✓ | ✗ | ✓ | ✓ |
| GAUCHE (ours) | ✓ | ✓ | ✓ | ✓ | ✓ | ✓ | ✓ |

# 6   Related Work

General-purpose GP and BO libraries do not cater for molecular representations. Likewise, general-purpose molecular machine learning libraries do not consider GPs and BO. Here, we review existing libraries, highlighting the niche GAUCHE fills in bridging the GP and molecular machine learning communities. The closest work to ours is FlowMO [72], which introduces a basic molecular GP library in the GPflow framework. In this project, we extend the scope of the library to a broader class of molecular representations (graphs), problem settings (BO) and applications (reaction optimisation and protein engineering). An overview of how GAUCHE fits into the existing open-source GP, BO and molecular machine learning stack is presented in Table 2.

**Gaussian Process Libraries**   GP libraries include GPy (Python) [73], GPflow (TensorFlow) [69, 70], GPyTorch (PyTorch) [26] and GPJax (Jax) [74] while examples of recent BO libraries include BoTorch (PyTorch) [27], Dragonfly (Python) [75], HEBO (PyTorch) [76] and Trieste (Tensorflow) [77]. The aforementioned libraries do not explicitly support molecular representations. Extending them to cover molecular representations, however, requires implementations of bespoke GP kernels for bit vector, string and graph inputs together with modifications to BO schemes to consider acquisition function evaluations over a discrete set of held-out molecules, a setting commonly encountered in virtual screening [78, 79].

**Molecular Machine Learning Libraries**   Molecular machine learning libraries include DeepChem [71], DGL-LifeSci [80] and TorchDrug [81]. DeepChem features a broad range of model implementations and tasks, while DGL-LifeSci focuses on graph neural networks. TorchDrug caters for applications including property prediction, representation learning, retrosynthesis, biomedical knowledge graph reasoning and molecule generation. However, none of the aforementioned libraries includes GP implementations. In terms of atomistic systems, DScribe [82] features, amongst other methods, the Smooth Overlap of Atomic Positions (SOAP) representation [83], which is typically used in conjunction with a GP model to learn atomistic properties. Automatic Selection And Prediction (ASAP) [84] also principally focuses on atomistic properties as well as dimensionality reduction and visualisation techniques for materials and molecules. Lastly, the Graphein library focuses on graph representations of proteins [49].

**Graph Kernel Libraries**   Graph kernel libraries include GraKel [59], graphkit-learn [85], graphkernels [86], graph-kernels [87], pykernels (https://github.com/gmum/pykernels) and ChemoKernel [88]. The aforementioned libraries focus on CPU implementations in Python. Extending graph kernel computation to GPUs has been noted as an important direction for future research [89]. In our work, we build on the GraKel library to construct GPyTorch-based GPs that can be trained on non-tensorial, graph-structured inputs. It is worth noting that GAUCHE extends the applicability of GPU-enabled GPs to general graph-structured inputs beyond just molecules and proteins.

**Molecular Bayesian Optimisation** BO over molecular space can be divided into two classes. In the first class, molecules are encoded into the latent space of a variational autoencoder (VAE) [4]. BO is then performed over the continuous latent space and queried molecules are decoded back to the original space. Much work on VAE-BO has focussed on improving the synergy between the surrogate model and the VAE [90, 5, 91, 92, 93, 94, 95, 96]. One of the defining characteristics of VAE-BO is that it enables the generation of new molecular structures. In the second class of methods, BO is performed directly over the original discrete space of molecules. In this setting it is not possible to generate new structures and so a candidate set of queryable molecules is defined. The inability to generate new structures however, is not a bottleneck to molecule discovery as the principle concern is how best to explore existing candidate sets. These candidate sets are also known as molecular libraries in the virtual screening literature [97]. To date, there has been little work on BO directly over discrete molecular spaces. In [56], the authors use a string kernel GP trained on SMILES to perform BO to select from a candidate set of molecules. In [98], an optimal transport kernel GP is used for BO over molecular graphs. In [99] a surrogate based on the Nadarya-Watson estimator is defined such that the kernel density estimates are inferred using BNNs. The model is then trained on molecular descriptors. Lastly, in [100] and [101] a BNN and a sparse GP respectively are trained on fingerprint representations of molecules. In the case of the sparse GP the authors select an ArcCosine kernel. It is a longstanding aim of the GAUCHE Project to compare the efficacy of VAE-BO against vanilla BO on real-world molecule discovery tasks.

**Chemical Reaction Optimisation** Chemical reactions describe how reactants transform into products. Reagents (catalysts, solvents, and additives) and reaction conditions heavily impact the outcome of chemical reactions. Typically the objective is to maximise the reaction yield (the amount of product compared to the theoretical maximum) [41], in asymmetric synthesis, where the reactions could result in different enantiomers, to maximise the enantiomeric excess [102], or to minimise the E-factor, which is the ratio between waste materials and the desired product [103]. A diverse set of studies have evaluated the optimisation of chemical reactions in single and multi-objective settings [103, 104]. [105] and [106] benchmarked reaction optimisation algorithms in low-dimensional settings including reaction conditions, such as time, temperature, and concentrations. [6] suggested BO as a general tool for chemical reaction optimisation and benchmarked their approach against human experts. [107] compared the yield prediction performance of different kernels and [108] the impact of various molecular representations. In all reaction optimisation studies above, the representations of the different categories of reactants and reagents are concatenated to generate the reaction input vector, which could lead to limitations if another type of reagent is suddenly considered. Moreover, most studies concluded that simple one-hot encodings (OHE) perform at least on par with more elaborate molecular representations in the low-data regime [6, 108, 109]. In GAUCHE, we introduce reaction fingerprint kernels, based on existing reaction fingerprints [46, 45] and work independently of the number of reactant and reagent categories.

# 7   Limitations

One potential limitation of GAUCHE is the focus on core implementations that are likely to remain robust across as many applied problems as possible which will enable the library to have the most impact. As such, there is less of a focus on bespoke GP implementations for more targeted problems [110, 111, 112, 113]. Nonetheless, we hope that GAUCHE will function as an active development platform for such implementations.

# 8   Conclusion

We have introduced GAUCHE, a library for Gaussian Processes in Chemistry, with the aim of providing a user-friendly and robust library of state-of-the-art uncertainty quantification and Bayesian optimisation tools that may hopefully be deployed for screening in laboratory settings. Our aim is to maintain a lean, well-tested and up-to-date codebase and invite community-driven contributions principally as pull requests in the form of notebooks that reflect the needs and considerations that researchers come across in practice. In this fashion, we may support more advanced features without bloating the codebase and increasing maintenance requirements.

# 9 Acknowledgements

BR and PS acknowledge support from the NCCR Catalysis (grant number 180544), a National Centre of Competence in Research funded by the Swiss National Science Foundation. ARJ is funded by a Biotechnology and Biological Sciences Research Council (BBSRC) DTP studentship (BB/M011194/1). GT acknowledges the support of the Natural Sciences and Engineering Research Council of Canada (NSERC), and the Vector Institute. FS-K is a postdoctoral fellow in the Eric and Wendy Schmidt AI in Science Postdoctoral Fellowship Program, a program by Schmidt Futures. AA-G acknowledges the generous support of Anders G. Frøseth, the Canadian Institute for Advanced Research (CIFAR), and the Canada 150 Research Chair program. LK acknowledges support from the University of Oxford's Clarendon Fund.

R-RG declares that this work was done exclusively at the University of Cambridge. Code/data was not created/accessed by Meta.

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

# Supplementary to:
# GAUCHE: A Library for Gaussian Processes in Chemistry

## A    Uncertainty Quantification Experiments

In Table 3 we present further uncertainty quantification results using the quantile coverage error (QCE) metric. Numerical errors were encountered with the WL kernel on the large lipophilicity dataset and the corresponding entry is left blank.

Table 3: UQ benchmark. QCE values ($\downarrow$) for 80/20 train/test split across 20 random trials.

| GP Model | | Dataset | | | |
|---|---|---|---|---|---|
| Kernel | Representation | Photoswitch | ESOL | FreeSolv | Lipophilicity |
| Tanimoto | fragprints | **0.019 ± 0.003** | 0.023 ± 0.002 | 0.023 ± 0.002 | 0.006 ± 0.002 |
| | fingerprints | 0.023 ± 0.003 | 0.022 ± 0.002 | 0.018 ± 0.003 | 0.006 ± 0.001 |
| | fragments | 0.025 ± 0.005 | 0.012 ± 0.002 | 0.014 ± 0.002 | 0.009 ± 0.002 |
| Scalar Product | fragprints | 0.033 ± 0.006 | 0.010 ± 0.002 | 0.017 ± 0.003 | 0.010 ± 0.001 |
| | fingerprints | 0.036 ± 0.006 | 0.014 ± 0.002 | 0.016 ± 0.002 | 0.009 ± 0.001 |
| | fragments | 0.027 ± 0.004 | 0.012 ± 0.003 | 0.021 ± 0.003 | 0.010 ± 0.001 |
| String | SMILES | 0.024 ± 0.003 | 0.016 ± 0.002 | 0.019 ± 0.003 | 0.005 ± 0.001 |
| WL Kernel (GraKel) | graph | 0.025 ± 0.007 | 0.011 ± 0.004 | 0.019 ± 0.009 | - |

## B    Chemical Reaction Yield Prediction Experiments

Further regression and uncertainty quantification experiments are presented in Table 4. The differential reaction fingerprint in conjunction with the Tanimoto kernel is the best-performing reaction representation.

Table 4: Chemical reaction regression benchmark. 80/20 train/test split across 20 random trials.

| GP Model | | Buchwald-Hartwig | | |
|---|---|---|---|---|
| Kernel | Representation | RMSE $\downarrow$ | $R^2$ score $\uparrow$ | QCE $\downarrow$ |
| Tanimoto | OHE | 7.94 ± 0.05 | 0.91 ± 0.001 | 0.011 ± 0.001 |
| | DRFP | **6.48 ± 0.45** | **0.94 ± 0.015** | 0.027 ± 0.002 |
| Scalar Product | OHE | 15.23 ± 0.052 | 0.69 ± 0.002 | 0.008 ± 0.001 |
| | DRFP | 14.63 ± 0.050 | 0.71 ± 0.002 | 0.010 ± 0.001 |
| RBF | RXNFP | 10.79 ± 0.049 | 0.84 ± 0.001 | 0.024 ± 0.001 |
| | | Suzuki-Miyaura | | |
| Tanimoto | OHE | 11.18 ± 0.036 | 0.83 ± 0.001 | 0.007 ± 0.001 |
| | DRFP | 11.46 ± 0.038 | 0.83 ± 0.001 | 0.019 ± 0.000 |
| Scalar Product | OHE | 19.91 ± 0.042 | 0.47 ± 0.003 | 0.012 ± 0.001 |
| | DRFP | 19.66 ± 0.042 | 0.52 ± 0.003 | 0.014 ± 0.001 |
| RBF | RXNFP | 13.83 ± 0.048 | 0.75 ± 0.002 | 0.007 ± 0.001 |

## C    Scaffold Split Experiments

To investigate how GP models behave on more challenging train/test splits, we have re-run parts of our experimental evaluation with 80-20 Bemis-Murcko [114] scaffold splits instead of random splits. As only the lipophilicity dataset exhibits sufficient scaffold diversity to perform this analysis (the skewness of the scaffold distribution in the others makes an 80-20 split impossible), the results in Table 5 focus on the predictive accuracy (RMSE) and calibration (NLPD) of GP models in this setting.

While this more challenging evaluation setup leads to slightly higher RMSEs and NLPDs, we note that one can observe the same trends as with random splits: Tanimoto-based GPs generally outperform Scalar Product ones, while string kernel-based GPs are better than both.

Table 5: Regression and uncertainty quantification experiments on scaffold splits.

| Kernel | Representation | RMSE ($\downarrow$) | NLPD ($\downarrow$) |
|---|---|---|---|
| Tanimoto | Fragprints | $0.86_{\pm 0.01}$ | $\mathbf{1.02}_{\pm \mathbf{0.04}}$ |
| | Fingerprints | $0.88_{\pm 0.01}$ | $1.12_{\pm 0.04}$ |
| | Fragments | $0.89_{\pm 0.01}$ | $2.10_{\pm 0.13}$ |
| Scalar Product | Fragprints | $0.89_{\pm 0.01}$ | $1.75_{\pm 0.08}$ |
| | Fingerprints | $0.95_{\pm 0.01}$ | $1.99_{\pm 0.09}$ |
| | Fragments | $1.00_{\pm 0.01}$ | - |
| String | SMILES | $\mathbf{0.82}_{\pm \mathbf{0.01}}$ | $1.08_{\pm 0.04}$ |

## D  Deep Probabilistic Models

There is a growing body of work applying deep learning to molecular property prediction [115]. Therefore in addition to evaluating GPs with varying shallow kernel functions, we repeat the regression experiments with a range of deep Bayesian models, varying both the network architecture and the Bayesian inference procedure. We evaluate the following models:

- **FC-BNN + VI** is a fully connected neural network with a single variational inference (VI) Bayesian layer with 100 nodes, followed by the rectified linear unit (ReLU) activation, and a final output layer. Trained with early stopping.

- **GNN-BNN + VI** utilises the same network and graph features as used for the graph embeddings [116] followed by a final VI Bayesian layer. Trained with early stopping.

- **CNN DKL GP** is the same approach and architecture used by [96] to predict molecular properties for Bayesian optimisation. Using SELFIES representations, a 1D CNN encoder is shared and trained jointly through a generative masked language model (MLM) head [47] and a discriminative deep kernel GP head [117].

- **CNN Ensemble** is a deep ensemble of 1D CNN networks, also implemented by [96], where each ensemble component uses the SELFIES molecule representation and is trained independently to minimize the MSE loss. Deep ensembles have been shown to provide high-fidelity approximations of Bayesian model averages relative to alternative approaches such as Laplace approximation or VI [118].

## E  Further Experiments

### E.1  Protein Fitness Prediction

We consider the task of protein fitness prediction where the fitness function (target label) takes the form of the melting point in degrees Celsius. We collate a dataset of 151 PETase protein sequences from values reported in [119, 120, 121]. PETases, recently discovered in 2016 [122], are a class of esterase enzymes which, via hydrolysis, catalyse the breakdown of polyethylene terephthalate (PET) plastic to monomeric mono-2-hydroxyethyl terephthalate (MHET). While PET plastics can require hundreds of years to degrade naturally, PETases are capable of degrading PET in a matter of days but their melting point is a key property of interest for deployed applications. Each sequence consists of an amino acid chain of length 290. We use the 'bag of amino acids' representation with a max n-gram value of 5 which gives rise to a count vector where each component represents the number of a given n-gram contained in the sequence. A max n-gram value of 5 means that all n-grams up to and including length 5 are included in the featurisation. We subsequently train a Tanimoto kernel GP

on the featurisation. The results are provided in Table 6 and indicate that the GP model obtains low generalisation error on the melting point prediction task.

Table 6: PETase melting point prediction experiment. 80/20 train/test split across 20 random trials.

| GP Model | | PETase Test Set | | |
| --- | --- | --- | --- | --- |
| Kernel | Representation | RMSE $\downarrow$ | MAE $\downarrow$ | $R^2$ score $\uparrow$ |
| Tanimoto | Bag of Amino Acids | $3.68 \pm 0.13$ | $2.54 \pm 0.10$ | $0.81 \pm 0.02$ |

## E.2   Molecular Preference Learning

In many optimisation problems, while it may be challenging to define a mathematical utility function, human feedback on pairwise comparisons can be leveraged to learn a latent utility function. In the machine learning literature, this observation has inspired work on preference learning [123, 124, 125, 126] where the goal is to learn a utility function $g(\mathbf{x})$ using binary preference data $r(\mathbf{x}_1, \mathbf{x}_2)$ obtained from human feedback on inputs $\mathbf{x}$. In the case of molecular design, there are many situations in which it is difficult to articulate a utility function [127, 128, 129] which has motivated recent work on human-in-the-loop preference learning [130, 131]. In this setting, pairwise preferences $r(\mathbf{x}_1, \mathbf{x}_2)$ are collected from a human chemist by presenting with many choices over pairs of molecules. In this section, we highlight a use-case for GAUCHE in GP-based preference learning [132]. Using the photoswitch dataset as a case study, we simulate noiseless pairwise comparisons from a human chemist on the E isomer $\pi - \pi^*$ transition wavelength (latent utility function). We train a Tanimoto kernel GP on the fragprints representation of the molecules, using a probit likelihood and the Laplace approximation [132]. The results are provided in Table 7. We report the Kendall-Tau rank correlation as the preference model learns ordinal rankings in place of absolute values of the transition wavelength. The results indicate that it is possible to learn accurate models of molecular properties through binary feedback alone.

Table 7: Photoswitch preference learning experiment. 80/20 train/test split across 20 random trials with 5000 training set pairwise comparisons. Kendall-Tau rank correlation (K-T) values reported.

| GP Model | | Photoswitch Dataset | |
| --- | --- | --- | --- |
| Kernel | Representation | K-T Train $\uparrow$ | K-T Test $\uparrow$ |
| Tanimoto | Fragprints | $0.95 \pm 0.001$ | $0.79 \pm 0.01$ |

## E.3   Ablation over the ECFP Radius Parameter

In addition to the experimental results presented in Section 5, we have performed an ablation study over the radius parameter of extended-connectivity fingerprints (ECFPs). Specifically, we trained Tanimoto-kernel GPs on the Photoswitch dataset using a series of five increasing radius parameters. In Table 8, we report the mean and standard error of the RMSE and NLPD over 50 different 80-20 train-test-splits.

Table 8: RMSE and NLPD of Tanimoto-kernel GPs trained on the Photoswitch dataset using a series of five increasing ECFP radius parameters.

| Fingerprint | RMSE ($\downarrow$) | NLPD ($\downarrow$) |
| --- | --- | --- |
| ECFP4 | **22.65$\pm$0.55** | **0.41$\pm$0.05** |
| ECFP6 | 23.50$\pm$0.55 | 0.47$\pm$0.05 |
| ECFP8 | 24.43$\pm$0.54 | 0.52$\pm$0.05 |
| ECFP10 | 25.17$\pm$0.54 | 0.56$\pm$0.04 |
| ECFP12 | 25.70$\pm$0.53 | 0.58$\pm$0.04 |

Intriguingly, these results show a strong negative correlation between the fingerprint radius and predictive performance. We hypothesize that this is caused by the fact that an expanding feature

space leads to lower and less informative Tanimoto similarity scores, making it more difficult to train generalisable models.

### E.4 Preferential Bayesian Optimisation

In many Bayesian optimisation problems, the acquisition strategy only requires rank-based preferences of candidates (as opposed to their absolute objective function values) to operate. Such an observation has motivated the field of Preferential Bayesian optimisation (PBO) [65, 66, 67, 68] where Bayesian optimisation is performed using binary preference data in lieu of the absolute values of the objective function. In Figure 5 we present the results of a Bayesian optimisation loop on the photoswitch dataset featuring a Tanimoto kernel GP surrogate with probit likelihood and the Laplace approximation [132], combined with the $EUBO - \zeta$ acquisition function from [68, 133]. The results indicate that it is possible to perform molecular Bayesian optimization using binary preference data alone.

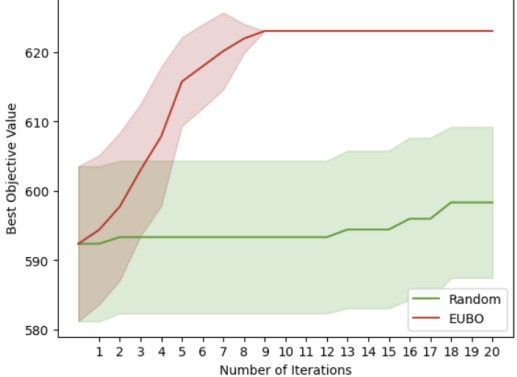

Figure 5: PBO results on the Photoswitch dataset using the $EUBO - \zeta$ acquisition function. 20 random trials reporting the 1.96 standard error bands. The surrogate is initialized with 98 molecules with 100 pairwise comparisons. 294 molecules are retained in a heldout set to be selected as part of the BO loop. A single molecule is selected on each iteration and an additional 100 pairwise comparisons over the augmented train set are recorded.

# F Coding Kernels in GAUCHE

We provide an example of the class definition for the Tanimoto kernel in GAUCHE below

```python
class TanimotoGP(ExactGP):
    def __init__(self, train_x, train_y, likelihood):
        super(TanimotoGP, self).__init__(train_x,
                                         train_y,
                                         likelihood)
        self.mean_module = ConstantMean()
        # We use the Tanimoto kernel to work with
        # molecular fingerprint representations
        self.covar_module = ScaleKernel(TanimotoKernel())

    def forward(self, x):
        mean_x = self.mean_module(x)
        covar_x = self.covar_module(x)
        return MultivariateNormal(mean_x, covar_x)
```

and an example definition of a black box kernel (where gradients with respect to hyperparameters and input labels are not required).

```python
class WLKernel(gauche.Kernel):
    def __init__(self):
        super().__init__()
        self.kernel = grakel.kernels.WeisfeilerLehman()

    @lru_cache(maxsize=3)
    def kern(self, X):
        return tensor(self.kernel.fit_transform(X.data))

class GraphGP(gauche.SIGP):
    def __init__(self, train_x, train_y, likelihood):
        super().__init__(train_x, train_y, likelihood)
        self.mean = ConstantMean()
        self.covariance = WLKernel()

    def forward(self, X):
        # X is a gauche.Inputs instance, with X.data
        # holding a list of grakel.Graph instances.
        mean = self.mean(zeros(len(X.data), 1))
        covariance = self.covariance(X)
        return MultivariateNormal(mean, covariance)
```

Importantly, GAUCHE inherits all the facilities of GPyTorch and GraKel allowing a broad range of of models to be defined on molecular inputs such as deep GPs, multioutput GPs and heteroscedastic GPs.

# G  Additional Bit and Count Vector Kernels

GAUCHE provides parallelisable and batch-GP-compatible implementations of the following similarity measures from [134] that are provably symmetric and positive semi-definite [135]. All kernels are defined for binary vectors $\mathbf{x}, \mathbf{x}' \in \{0, 1\}^d$ for $d \geq 1$, where $n$ represents the number of common zeros between $\mathbf{x}$ and $\mathbf{x}'$ and $|| \cdot ||$ is the Euclidean norm.

**Braun-Blanquet Kernel**

$$k_{\text{Braun-Blanquet}}(\mathbf{x}, \mathbf{x}') := \sigma_f^2 \cdot \frac{\langle \mathbf{x}, \mathbf{x}' \rangle}{\max(\|\mathbf{x}\|, \|\mathbf{x}'\|)}.$$

**Dice Kernel**

$$k_{\text{Dice}}(\mathbf{x}, \mathbf{x}') := \sigma_f^2 \cdot \frac{2 \cdot \langle \mathbf{x}, \mathbf{x}' \rangle}{\|\mathbf{x}\| + \|\mathbf{x}'\|}.$$

**Faith Kernel**  [136]

$$k_{\text{Faith}}(\mathbf{x}, \mathbf{x}') := \sigma_f^2 \cdot \frac{2 \cdot \langle \mathbf{x}, \mathbf{x}' \rangle + n}{2d}.$$

**Forbes Kernel**  [137, 138]

$$k_{\text{Forbes}}(\mathbf{x}, \mathbf{x}') := \sigma_f^2 \cdot \frac{d \cdot \langle \mathbf{x}, \mathbf{x}' \rangle}{\|\mathbf{x}\| + \|\mathbf{x}'\|}.$$

**Intersection Kernel**

$$k_{\text{Intersection}}(\mathbf{x}, \mathbf{x}') := \sigma_f^2 \cdot (\langle \mathbf{x}, \mathbf{x} \rangle + \langle \mathbf{x}', \mathbf{x}' \rangle).$$

**MinMax Kernel**

$$k_{\text{MinMax}}(\mathbf{x}, \mathbf{x}') := \sigma_f^2 \cdot \frac{\|\mathbf{x}\| + \|\mathbf{x}'\| - \|\mathbf{x} - \mathbf{x}'\|}{\|\mathbf{x}\| + \|\mathbf{x}'\| + \|\mathbf{x} - \mathbf{x}'\|}.$$

**Otsuka Kernel**

$$k_{\text{Otsuka}}(\mathbf{x}, \mathbf{x}') := \sigma_f^2 \cdot \frac{\langle \mathbf{x}, \mathbf{x}' \rangle}{\sqrt{\|\mathbf{x}\| + \|\mathbf{x}'\|}}.$$

**Rand Kernel**

$$k_{\text{Rand}}(\mathbf{x}, \mathbf{x}') := \sigma_f^2 \cdot \frac{\langle \mathbf{x}, \mathbf{x}' \rangle + n}{d}.$$

**Rogers-Tanimoto Kernel**

$$k_{\text{Rogers-Tanimoto}}(\mathbf{x}, \mathbf{x}') := \sigma_f^2 \cdot \frac{\langle \mathbf{x}, \mathbf{x}' \rangle + n}{2 \cdot \|\mathbf{x}\| + 2 \cdot \|\mathbf{x}'\| - 3 \cdot \langle \mathbf{x}, \mathbf{x}' \rangle + n}.$$

**Russell-Rao Kernel**

$$k_{\text{Russell-Rao}}(\mathbf{x}, \mathbf{x}') := \sigma_f^2 \cdot \frac{\langle \mathbf{x}, \mathbf{x}' \rangle}{d}.$$

**Sorgenfrei Kernel**

$$k_{\text{Sorgenfrei}}(\mathbf{x}, \mathbf{x}') := \sigma_f^2 \cdot \frac{\langle \mathbf{x}, \mathbf{x}' \rangle^2}{\|\mathbf{x}\| + \|\mathbf{x}'\|}.$$

**Sokal-Sneath Kernel**  [139]

$$k_{\text{Sokal-Sneath}}(\mathbf{x}, \mathbf{x}') := \sigma_f^2 \cdot \frac{\langle \mathbf{x}, \mathbf{x}' \rangle}{2 \cdot \|\mathbf{x}\| + 2 \cdot \|\mathbf{x}'\| - 3 \cdot \langle \mathbf{x}, \mathbf{x}' \rangle}.$$

