# OpenReview forum: "GAUCHE: A Library for Gaussian Processes in Chemistry"
_NeurIPS.cc/2023/Conference — NeurIPS 2023 poster_

### Official Review · Reviewer_JWPZ · 2023-06-20

**Soundness:** 2 fair
**Presentation:** 2 fair
**Contribution:** 3 good
**Rating:** 6
**Confidence:** 3

**Summary:**

Gaussian processes are widely used for black-box optimization when data is scarce. On the other hand, effectively representing molecules, proteins, and chemical reactions is a dedicated research area in molecular machine learning.

Although separate tools exist to address these two challenges, this paper introduces a novel library that integrates both. By doing so, it enables chemists without extensive knowledge of Bayesian optimization to harness its benefits.

**Strengths:**

The library's objective is clearly defined and holds potential for the chemistry and machine learning research communities. The persuasive comparison with existing work in Section 5 strengthens its position. Sections 2.3 and 2.4 provide thorough explanations of molecular and chemistry reactions.

**Weaknesses:**

I find the objective of the paper somewhat unclear. Here is my understanding and suggested improvements:

* The paper's main aim, as summarized in Section 5, is to create a unified library called GAUCHE that combines chemistry libraries (for molecule/chemical reaction representation) with Bayesian optimization libraries. Such libraries exist separately, GAUCHE aims to unify them.
* Sections 2 and 3 effectively describe the library's provided representations and kernels.
* However, I fail to grasp the purpose of the experiments described in Section 4. The message seems to be that "the library works and enables Bayesian optimization on various chemistry benchmarks." This leaves room for improvement: what sets GAUCHE apart? Why should users choose it over manually combining a molecular representation with a GP library? Section 5 partially answers these questions – and should be presented earlier in the paper, in my opinion.
* Overall, it appears that the library does not offer new theoretical insights or algorithms. While this is not necessarily a problem (as GAUCHE fills a relevant niche, allowing non-GP-expert chemists to utilize state-of-the-art black-box optimization for chemical reactions), **the paper should clearly emphasize the library's strengths**. These may include (a) ease of use, (b) a modular code structure enabling user extension, or (c) superior performance on public benchmarks. In that spirit, a helpful addition would be a table comparing different libraries, with the different libraries (containing GAUCHE) in rows and the different features (GP, molecular representation, etc) in columns, and cells indicating whether the feature is implemented or not.

**Questions:**

* `l. 60`, the covariance $\sigma_y$ is not introduced. I understand that this assumes a Bayesian linear model of the form $f({\bf x}) = \phi({\bf x})^\top {\bf w} + \sigma_y \epsilon$ where $\epsilon$ follows a standard normal distribution and $K({\bf x}, {\bf x}') = \phi({\bf x})^\top \phi ({\bf x}')$, as in [1, Eq. 2.21]. I am not familiar with the BO literature, and perhaps it's the only model considered, but it would be worth mentioning that you are considering a Bayesian linear model, along with a proper citation to justify the formula.
* `l. 68`: again, adding a reference e.g. [1, Eq. 5.8] seems necessary to justify the NLML formula.
* Fig. 1 is not informative for someone who does not have GP or chemistry background. After reading sec. 2.{3, 4, 5}, we understand that the 3rd row show the possible representation for the different applications considered in the 2nd row (even though "SMARTS" in not described in sec. 2. 4). In any case, the figure would benefit from a more detailed caption, so that the message it is trying to convey appears clearly.
* Table 1 and 2: why the font is so small? It would be convenient to have the same font size as the rest of the document.
* `l. 227`, I don't know the 3 metrics for measuring the quality of the uncertainty estimates. A word of introduction explaining what they measure, what are their strength and limitation would be welcomed here.

**Related literature.**

* You may be interested in [2] which provides an convolutional kernel network for graph-structured data.

**Typos.**

* `l. 55`: $m(\mathbf{x'})$ instead of $m(\mathbf{x})$
* `l. 80`: extra "the"

**References.**

[1] Rasmussen and Williams - 2006 - Gaussian processes for machine learning

[2] Convolutional Kernel Networks for Graph-Structured Data – 2020 – Dexiong Chen, Laurent Jacob, Julien Mairal

**Limitations:**

The limitations identified by the authors regarding specific algorithms for certain problems are noteworthy but can be left for future research. Aside from that, my main concerns are summarized in the "Weaknesses" section mentioned previously.

---

> ### Author Rebuttal · Authors · 2023-08-10
>
> &nbsp;
>
> Thank you for taking the time to review our manuscript and for providing detailed, helpful and constructive feedback. We were happy to see that you appreciate the practical usefulness of a well-designed and easy-to-use library that enables scientific experts to make use of Bayesian optimisation in low-data regimes.
>
> The main concern you raised in your review relates to the way we present the objectives and strengths of our work and we sincerely appreciate the feedback. We will try to address your points below.
>
> &nbsp;
>
> ## __Clarifying the Added Value of GAUCHE__ ##
>
> &nbsp;
>
> > The paper’s main aim, as summarized in Section 5, is to create a unified library called GAUCHE that combines chemistry libraries (for molecule/chemical reaction representation) with Bayesian optimization libraries. Such libraries exist separately, GAUCHE aims to unify them.
>
> &nbsp;
>
> As you correctly stated, our main aim is to create an open and unified community resource that makes it as easy as possible to combine current (and future) state-of-the-art molecular representations and kernels with Gaussian Process and Bayesian Optimization libraries. The central motivation behind this approach is to create a public repository that enables expert chemists and materials scientists with little background in GPs or BO to make use of state-of-the-art black-box optimization techniques, as well as to minimize the time and effort spent on re-implementing redundant optimization pipelines.
>
> &nbsp;
>
> > However, I fail to grasp the purpose of the experiments described in Section 4. (…) What sets GAUCHE apart? Why should users choose it over manually combining a molecular representation with a GP library?
>
> &nbsp;
>
> The purpose of the experiments in Section 4 (and the corresponding Jupyter Notebook tutorials) is to provide prospective users with a convincing demonstration that GAUCHE presents a modular and user-friendly platform for the rapid exploration and prototyping of different molecular representations and similarity kernels to establish which setup (if any) works best for a given application.
>
> &nbsp;
>
> > (…) the paper should clearly emphasize the library’s strengths. These may include (a) ease of use, (b) a modular code structure enabling user extension, or (c) superior performance on public benchmarks.
>
> &nbsp;
>
> We will make sure to refine these points in Section 5 and to also clearly state them in both the Abstract and Introduction of the paper.
>
> &nbsp;
>
> > In that spirit, a helpful addition would be a table comparing different libraries (…)
>
> &nbsp;
>
> Thank you for the suggestion. We agree that clearly stating the value that GAUCHE adds over existing libraries by summarizing Section 5 into an intuitive table would help to further clarify the points we make above. We have included a draft of this table below and a more polished version in the pdf submitted with our general response.
>
> &nbsp;
>
> | Library  | Gaussian Processes | Bayesian Optimisation | Molecular Representations | Chemistry Tutorials | Graph Kernels | Bit Vector Kernels | String Kernels |
> |----------|--------------------|-----------------------|---------------------------|---------------------|---------------|--------------------|----------------|
> | GPyTorch | ✓                  | ✗                     | ✗                         | ✗                   | ✗             | ✗                  | ✗              |
> | GPflow   | ✓                  | ✗                     | ✗                         | ✗                   | ✗             | ✗                  | ✗              |
> | BoTorch  | ✓                  | ✓                     | ✗                         | ✗                   | ✗             | ✗                  | ✗              |
> | DeepChem | ✗                  | ✗                     | ✓                         | ✓                   | ✗             | ✗                  | ✗              |
> | GraKel   | ✗                  | ✗                     | ✗                         | ✗                   | ✓             | ✗                  | ✗              |
> | FlowMO   | ✓                  | ✗                     | ✓                         | ✓                   | ✗             | ✓                  | ✓              |
> | GAUCHE   | ✓                  | ✓                     | ✓                         | ✓                   | ✓             | ✓                  | ✓              |
>
>
> &nbsp;
>
> ## __Questions__ ##
>
> &nbsp;
>
> 1. Great spot on the $\sigma_y$ term. This indeed represents a coefficient for additive Gaussian noise and we will update the manuscript to include this definition.
> 2. Many thanks for the reference! We have opened a GitHub issue to introduce convolutional kernel networks.
>
> &nbsp;
>
> ## __Summary__ ##
>
> &nbsp;
>
> We hope that these points clarify the objective of the paper and address the presentational concerns you raised.
>
> Please let us know if you have any further questions!
>
> &nbsp;
>
> Sincerely,
>
> The Authors
>
> &nbsp;

---

> > ### Comment · Reviewer_JWPZ · 2023-08-11
> >
> > Thank to the authors for their rebuttal. The table comparing GAUCHE to other frameworks is compelling. I believe that any libraries that streamline the utilization of advanced machine learning tools are valuable.
> >
> > **I am willing to raise my rating from 5 to 6**. I still have reservations about fully assessing the library's contribution.

---

> > > ### Author Response · Authors · 2023-08-12
> > > **On GAUCHE's Contribution**
> > >
> > > &nbsp;
> > >
> > > Thank you for the quick response! We are happy to hear that you found the table compelling and that you appreciate the value of a library that lowers the barrier for applying advanced molecular machine learning tools in practical research settings.
> > >
> > > &nbsp;
> > >
> > > > I still have reservations about fully assessing the library’s contribution.
> > >
> > > &nbsp;
> > >
> > > To address your remaining concern and further clarify the contributions of our work, we would like to expand on the points in our initial rebuttal to:
> > >
> > > &nbsp;
> > >
> > > 1. Highlight some of the technical innovations that were necessary to extend the existing open-source GP/BO stack to work for discrete kernels that operate over molecular representations (i.e. bit vectors and graphs).
> > >
> > > 2. Outline the concrete impact our work has had by listing instances in which GAUCHE has already been applied by other researchers in a range of practical settings.
> > >
> > > &nbsp;
> > >
> > > ## __Substantial Technical Contributions__ ##
> > >
> > > &nbsp;
> > >
> > > One important and major limitation of existing GP frameworks is that they are built with continuous data in $\mathbb{R}^d$
> > > in mind. For instance, the kernel base class of GPyTorch assumes that custom kernel sub-classes are based on Euclidean distance metrics. In GAUCHE, we provide a parallelizable and batch-GP-compatible alternative to this base class that can be easily extended to implement arbitrary bit and count vector kernels. As another example, we would like to point out that current GPU-enabled GP libraries do not natively support non-tensorial inputs, necessitating a substantial amount of engineering work to extend them to graph-structured input spaces. The resulting SIGP class is, to the best of our knowledge, the first open-source GP implementation that enables GPU-accelerated and autodiff-based end-to-end learning over graph kernel hyperparameters, including all kernels in the GraKel library. While certain kernels - such as the Weisfehler-Lehman (WL) kernel - have concrete feature functions that could be fit directly by a GP, many others - such as the random walk and shortest path kernel - don’t, which was our motivation for designing a more general wrapper framework.
> > >
> > > &nbsp;
> > >
> > > ## __Real-World Impact__ ##
> > >
> > > &nbsp;
> > >
> > > By combining these technical innovations with a range of easy-to-adapt data loaders, featurization functions and notebook tutorials, we made it as easy as possible to integrate molecular GP and BO models into real-world research workflows. Specifically, we would like to mention that GAUCHE has already successfully contributed to a range of academic and industrial research efforts. We are currently aware of at least three application domains in which GAUCHE has featured as a core component of published work:
> > >
> > > &nbsp;
> > >
> > > 1. Additive screening for chemical reaction optimization.
> > > 2. Catalyst discovery
> > > 3. Self-driving laboratories
> > >
> > > &nbsp;
> > >
> > > Additionally, GAUCHE has been a core component in enabling novel Bayesian optimization methodologies to be evaluated on molecular datasets. Published work in this direction has included:
> > >
> > > &nbsp;
> > >
> > > 1. The evaluation of a novel multiobjective Bayesian optimization scheme on the task of identifying molecules with favourable cell permeability for drug delivery.
> > > 2. The evaluation of a novel method featuring Bayesian quadrature on a) the task of identifying molecules with anti-malarial properties and b) the task of identifying molecules with promising solvation capabilities for use in lithium-ion battery electrolytes.
> > >
> > > &nbsp;
> > >
> > > We have consulted the AC and SAC on the best way to provide these references without compromising the double-blind review process and are currently waiting to hear back.
> > >
> > > &nbsp;
> > >
> > > ## __Summary__ ##
> > >
> > > &nbsp;
> > >
> > > We hope that these clarifications are helpful in allowing you to fully assess the contributions of our work and are more than happy to provide additional details and answer any follow-up questions!
> > >
> > > &nbsp;
> > >
> > > Sincerely,
> > >
> > > The Authors

---

### Official Review · Reviewer_rHxF · 2023-06-25

**Soundness:** 3 good
**Presentation:** 3 good
**Contribution:** 2 fair
**Rating:** 5
**Confidence:** 4

**Summary:**

The authors discuss Molecular, Reaction, and Protein Representations and provide a unified framework for these models. Python's GPyTorch library is used to train the Gaussian processes. The authors define certain kernels for Gaussian processes to fit and perform several experiments to evaluate the performance. The paper also contains a brief overview of  Gaussian processes and Bayesian optimization.

**Strengths:**

The authors properly analyze the related kernel functions used in chemistry. Indeed, they provide an overview of the applications and representations available in the GAUCHE library. The library seems easy to follow and can adapt to proteins, molecules, and chemical reactions.

**Weaknesses:**

The major weakness is the contribution. The training task uses GPyTorch, a powerful library in Python for Gaussian processes.
The main contribution of the paper is to provide some important kernels in chemistry. It seems some of the kernels have been developed in other libraries or their scripts are available on the internet. The theory part of the paper is not rich enough. Section 3 only explains the relevant kernels and does not provide novel ideas or solutions for the methodological challenges. The Gaussian process methods and kernels are already there without new inventions, and the paper does not describe how to adapt them to chemistry problems. It would be great if the authors describe the major benefits of the proposed solution compared to the other libraries related to the Gaussian processes, except the kernels.

**Questions:**

What are the major limitations of the available packages in R/Python  so that they can not be used in the chemistry problems and how this package addresses them? If the kernels are omitted, what is the main advantage of the library compared to the others?

**Limitations:**

The authors addressed the limitations of the paper.

---

> ### Author Rebuttal · Authors · 2023-08-10
>
> &nbsp;
>
> Thank you for taking the time to review our manuscript and for providing valuable and helpful feedback. We were happy to see that you appreciated the thorough treatment of the representations, kernels and applications we cover in our work.
>
> The main concern you raised in your review is the novelty of our contributions, which we aim to clarify below.
>
> &nbsp;
>
> ## __Clarifying the Added Value of GAUCHE__ ##
>
> &nbsp;
>
> > It would be great if the authors describe the major benefits of the proposed solution compared to the other libraries related to the Gaussian processes, except the kernels.
>
> &nbsp;
>
> The main contribution of our work is to create an open and unified community resource that makes it as easy as possible to combine current (and future) state-of-the-art molecular representations and kernels with existing Gaussian Process and Bayesian Optimization infrastructure. The central motivation behind this approach is to create a public repository that enables expert chemists and materials scientists with little background in GPs or BO to make use of state-of-the-art black-box optimization techniques, as well as to minimize the time and effort spent on re-implementing redundant optimization pipelines.
>
> We refer to Section 5 of the manuscript for a thorough review of prior art that aims to convey the added value that GAUCHE provides over existing libraries. For enhanced clarity, we have summarised this comparison as a table in the general comment above as well as in the attached pdf.
>
> &nbsp;
>
> > The main contribution of the paper is to provide some important kernels in chemistry. It seems some of the kernels have been developed in other libraries or their scripts are available on the internet. (…) Section 3 only explains the relevant kernels and does not provide novel ideas or solutions for the methodological challenges.
>
> &nbsp;
>
> While it is correct that we mostly build on kernels that are well-established in the literature, we would like to point out that
>
> 1. Discrete kernels are not straight-forwardly compatible with the design assumptions of GPyTorch/BOTorch (see below for more), and that a unified and open-source repository of compatible implementations is very helpful
> 2. That many of these kernels have never been used in the context of GP regression
> 3. That fewer still have been used for Bayesian optimization - especially in the context of chemistry, materials science and structural biology.
>
> &nbsp;
>
> We strongly agree that the development of novel and more performant kernels and representations is an important area of future research, but would like to point out that this is an orthogonal gap in the literature to the one we aim to address with this manuscript - though one that could strongly benefit from the robust foundation we aim to provide with GAUCHE.
>
> &nbsp;
>
> > The Gaussian process methods and kernels are already there without new inventions, and the paper does not describe how to adapt them to chemistry problems.
>
> &nbsp;
>
> We would like to contest the claim that we do not describe how to adapt our framework to chemistry problems, as we put significant effort into providing a range of easy-to-adapt Jupyter Notebook tutorials that demonstrate how to use GAUCHE for various real-world tasks in medicinal chemistry and reaction optimization. Going even further, we would like to point out that the library has already been used in a range of real-world production settings. We are unsure of how to link to these examples without violating the double-blindness of the review process and are currently liaising with the AC.
>
> &nbsp;
>
> ## __Questions__ ##
>
> &nbsp;
>
> > What are the major limitations of the available packages in R/Python so that they can not be used in the chemistry problems and how this package addresses them?
>
> &nbsp;
>
> One important and major limitation of existing GP frameworks is that they are built with continuous data in $\mathbb{R}^d$ in mind. For instance, the kernel base class of GPyTorch assumes that custom kernel sub-classes are based on Euclidean distance metrics. In GAUCHE, we provide a parallelizable and batch-GP-compatible alternative to this base class that can be easily extended to implement arbitrary bit and count vector kernels. As another example, we would like to point out that the associated GP optimization utilities only work for tensor-valued input spaces and required substantial adjustment to work on graph-structured inputs.
>
> &nbsp;
>
> ## __Summary__ ##
>
> &nbsp;
>
> We hope that these points clarify the objective of the paper and address the presentational concerns you raised. Please let us know if you have any further questions!
>
> &nbsp;
>
> Sincerely,
>
> The Authors
>
> &nbsp;

---

> > ### Comment · Reviewer_rHxF · 2023-08-21
> >
> > Thanks a lot for the author's comprehension answers to the concerns and questions. I checked the rebuttal and also other reviewers' comments. Also, the list of papers that used this library was useful.
> > Although I still have concerns about the contribution and theoretical novelty of this work, I want to raise my rating from 4 to 5.

---

> > > ### Author Response · Authors · 2023-08-22
> > > **Thank You for the Response and Feedback**
> > >
> > > &nbsp;
> > >
> > > Many thanks once again for taking the time to review the paper and offer feedback!
> > >
> > > &nbsp;
> > >
> > > Sincerely,
> > >
> > > The Authors

---

### Official Review · Reviewer_oVt5 · 2023-06-30

**Soundness:** 3 good
**Presentation:** 4 excellent
**Contribution:** 3 good
**Rating:** 7
**Confidence:** 3

**Summary:**

This article presents a library for Gaussian process-based inference with a special focus on chemistry applications. At heart, the library contains two classes of objects: kernel, and data loaders. The article introduces Gaussian processes and chemistry-specific kernels and discusses the interfacing of the library with other frameworks, in particular relating to Bayesian optimisation.

**Strengths:**

The proposed library seems to fill a gap in the open-source GP stack, providing specialised building blocks for use in chemistry. The code cleanliness is rather high and the code is well-tested.

The paper is fairly clear (see however my questions in the weaknesses), and provides a reasonable introduction to the problem and to the GP literature around it, some illustration of the superior performance of GPs (compared to Grakel) is adequately shown. The relation to prior works and existing solutions is also well documented.

**Weaknesses:**

As it stands, there is no clear indication in the article that the library has had any impact on the practice of data-driven chemistry. If possible, the authors should include examples of real-life uses of GAUCHE, and if there is none, it then would mean that the library is likely not mature enough to warrant a full-size article (although it would likely be a good workshop paper).

I have found it hard to understand which "20+ bespoke" kernels were in fact implemented. For instance, the graph kernels are described in the article, but the code provided shows nothing under "gauche/kernels/graph_kernels". Similarly, I cannot count 20 kernels in the code and would like for the authors to specifically list these.

The future of the library is also not mentioned: what is the governance model and what are the next steps? Is it to implement more kernels? When new kernels are implemented, then what? As it stands it seems that the authors suggest they will "wait and see", depending on the feedback they receive from the practitioners. This makes me believe that publishing an article on the library itself is then fairly premature.

**Questions:**

As discussed in the weaknesses, my main reasons for the negative rating are
(i) that the supported methods are not well documented within the article
(ii) the future of the library is really unclear: for instance
> We seek to further grow our userbase and solicit feedback from laboratory practitioners on the most common use-cases for BO and GP modelling in molecular discovery campaigns.

is an admission of stale development and of an unclear vision for the future of the library.
Some clarity on this would be much appreciated.


A remark on my confidence level: I am by no means in the position to judge the relevancy of introducing a package for chemistry-specialised kernels and do not really understand the end applications. I will therefore fully defer to other reviewers when it comes to this.

---

> ### Author Rebuttal · Authors · 2023-08-10
>
> &nbsp;
>
> Thank you for taking the time to review our manuscript and for providing helpful and constructive feedback. We were happy to see you emphasize how our library complements the current open-source molecular machine learning stack and acknowledge the high quality of our code and tests.
>
> The concerns you raised in your review relate to examples of real-life use-cases, our open-source governance model, and the claims we make regarding the number of available kernels. We will address each of these points in turn below.
>
> &nbsp;
>
> ## __Examples of Real-World Use-Cases__ ##
>
> &nbsp;
>
> > If possible, the authors should include examples of real-life uses of GAUCHE
>
> &nbsp;
>
> This is a great point and central to our motives for introducing GAUCHE. We are eager to share at least four instances (for which public references are available) in which GAUCHE has been utilized in real-world research and production settings. As stated in the general comment, we will liaise with the AC/SAC to determine the best way to provide these references!
>
> &nbsp;
>
> ## __Plans for Future Development__ ##
>
> &nbsp;
>
> > the future of the library is really unclear
>
> &nbsp;
>
> Pending the advice of the AC on the double-blind policy, if we are able to share some recent applications of GAUCHE this should give some flavour of the user-inspired extensions we have recently implemented! Additionally, in the attached pdf we have included a new SOTA performance on the photoswitch benchmark motivated by **reviewer rK4e** suggestion to investigate additional fingerprints/descriptors.
>
> &nbsp;
>
> ## __Number of Available Kernels__ ##
>
> &nbsp;
>
> > I have found it hard to understand which "20+ bespoke" kernels were in fact implemented.
>
> &nbsp;
>
> For the graph kernels alone, the total count subsumes the total number of kernels available in the GraKel library (18). In the "**external_graph_kernels.ipynb**" notebook we showcase the application of our wrapper around GraKel, the SIGP class which allows any kernel from GraKel to be used as a component of a PyTorch-based Gaussian process. Since GPU-enabled GP libraries do not natively support non-tensorial inputs, the SIGP class required a substantial amount of engineering work, but enables autodiff-based, end-to-end learning over the kernel hyperparameters of the GraKel library. Certain kernels such as the Weisfehler-Lehman (WL) kernel have concrete feature functions, and so a WL kernel-GP can be implemented by fitting a linear kernel to those features. However, given that many of GraKel's kernels do not possess distinctive feature functions, we decided to implement a more general wrapper framework. Additionally, we are in the process of integrating an additional suite of bit vector kernels in the coming week.
>
> &nbsp;
>
> ## __Summary__ ##
>
> &nbsp;
>
> We hope that our additional clarifications and discussion address all of your questions and concerns.
>
> Please let us know if you have any further questions!
>
> &nbsp;
>
> Sincerely,
>
> The Authors
>
> &nbsp;

---

> > ### Comment · Reviewer_oVt5 · 2023-08-14
> > **Rebuttal acknowledgement**
> >
> > Thank you for the clear response (both to me and other reviewers). Because sharing the examples is pending AC approval/guidelines, I will refrain from updating my score just yet. As it stands I am however inclined to increase it substantially (subject to the examples provided)
> >
> > > For the graph kernels alone, the total count subsumes the total number of kernels available in the GraKel library (18)
> >
> > I believe this should be very clearly stated rather than the fact that you provide 22+ kernels. The fact that your library immediately extends another one with no real overhead is a good thing.
> >
> > > Examples of Real-World Use-Cases
> >
> > I am eager to see these.
> >
> > > Plans for Future Development
> >
> > I would highly recommend making the short and long-term plans clearer in the conclusion of the article: if anything to prevent the kind of reaction I personally had thinking it was "an admission of stale development and of an unclear vision for the future of the library".

---

> > > ### Author Response · Authors · 2023-08-16
> > > **Many Thanks for the Prompt Response and Additional Feedback!**
> > >
> > > &nbsp;
> > >
> > > Thank you for your response! We are happy to hear that you found our clarifications helpful and sincerely appreciate the additional feedback.
> > >
> > > &nbsp;
> > >
> > > ## __Additional Kernels__ ##
> > >
> > > &nbsp;
> > >
> > > > I believe this should be very clearly stated rather than the fact that you provide 22+ kernels. The fact that your library immediately extends another one with no real overhead is a good thing.
> > >
> > > &nbsp;
> > >
> > > Thank you. We agree that this should indeed be stated more clearly in the paper and we will ensure that the graph kernel section (3.3) is adjusted accordingly. Motivated by your comment, we have additionally implemented 12 new bit vector kernels based on the following similarity measures from [1] and [2]:
> > >
> > > &nbsp;
> > >
> > > 1. Dice
> > > 2. MinMax
> > > 3. Sokal-Sneath
> > > 4. Russell and Rao
> > > 5. Sogenfrei
> > > 6. Forbes
> > > 7. Intersection
> > > 8. Faith
> > > 9. Otsuka
> > > 10. Rogers-Tanimoto
> > > 11. Braun-Blanquet
> > > 12. Rand
> > >
> > > &nbsp;
> > >
> > > All of these measures yield provably symmetric positive-definite kernels [3] and we provide parallelisable and batch-GP-compatible implementations with associated unit tests. We have also evaluated each of these kernels on the Photoswitch dataset, finding that they often slightly outperform the more standard Jaccard-Tanimoto kernel - particularly in the case of the Sorgenfrei kernel (number 5).
> > >
> > > &nbsp;
> > >
> > > ## __Real-World Use-Cases__ ##
> > >
> > > &nbsp;
> > >
> > > We are still waiting to hear back from the AC/SAC regarding the best way to share external references and will follow up with them shortly.
> > >
> > > &nbsp;
> > >
> > > > I am eager to see these.
> > >
> > > &nbsp;
> > >
> > > To provide you with as much information as possible whilst we wait, we think that it is safe to share the specific application domains in which GAUCHE has featured as a core component of published work. These include:
> > >
> > > &nbsp;
> > >
> > > 1. Additive screening for chemical reaction optimization.
> > > 2. Discovery and optimization of novel catalysts.
> > > 3. Self-driving laboratories and computational experiment planning.
> > >
> > > &nbsp;
> > >
> > > Additionally, GAUCHE has been a core component in enabling novel Bayesian optimization methodologies to be evaluated on molecular datasets. Published work in this direction includes:
> > >
> > > &nbsp;
> > >
> > > 1. The evaluation of a novel multiobjective Bayesian optimization scheme on the task of identifying molecules with favourable cell permeability for drug delivery.
> > > 2. The evaluation of a novel method featuring Bayesian quadrature on a) the task of identifying molecules with anti-malarial properties and b) the task of identifying molecules with promising solvation capabilities for use in lithium-ion battery electrolytes.
> > >
> > > &nbsp;
> > >
> > > We hope that these additional elaborations are helpful and provide useful context to inform your decision-making process.
> > >
> > > &nbsp;
> > >
> > > ## __Governance Model__ ##
> > >
> > > &nbsp;
> > >
> > > > I would highly recommend making the short and long-term plans clearer in the conclusion of the article
> > >
> > > &nbsp;
> > >
> > > Thank you. We agree that the conclusion should more clearly convey our short-term development and long-term governance plans and will adjust it accordingly. Regarding the latter, our aim is to maintain a lean, well-tested, and up-to-date main codebase. Following the maintenance model of GPflow, which has proved successful, we aim to invite community-driven contributions principally as PRs in the form of notebooks (as opposed to extensions to the main codebase) that reflect the needs and considerations that researchers come across in practice. In this fashion, we may support more advanced features without bloating the codebase and increasing maintenance requirements.
> > >
> > > &nbsp;
> > >
> > > We thank you again for your feedback and are happy to answer any further questions!
> > >
> > > &nbsp;
> > >
> > > Sincerely,
> > >
> > >
> > > The Authors
> > >
> > > &nbsp;
> > >
> > > ## __References__ ##
> > >
> > > &nbsp;
> > >
> > > [1] Choi, Seung-Seok, Sung-Hyuk Cha, and Charles C. Tappert. “A Survey of Binary Similarity and Distance Measures.” Journal of Systemics, Cybernetics and Informatics 8.1 (2010): 43-48.
> > >
> > > [2] Todeschini, R., D. Ballabio, and V. Consonni. “Distances and Similarity Measures in Chemometrics and Chemoinformatics.” Encyclopedia of Analytical Chemistry. RA Meyers, 2020. 1-40.
> > >
> > > [3] Nader, Rafic, et al. “On the Positive Semi-Definite Property of Similarity Matrices.” Theoretical Computer Science 755 (2019): 13-28.
> > >
> > > &nbsp;

---

> > > > ### Author Response · Authors · 2023-08-20
> > > > **Follow-up on References of Real-World Use Cases**
> > > >
> > > > Thank you again for taking the time to respond to our rebuttal.
> > > >
> > > > &nbsp;
> > > >
> > > > Following up on the best way to share references of real-world use cases, we have reached out to the Program Chairs, whose recommendation was to
> > > >
> > > > > provide the references directly to the AC, which can confirm for the reviewer that the library is being used.
> > > >
> > > > We have reached out to the AC again and will provide them with these references as instructed. In the meantime, we hope that the more detailed descriptions of the specific application domains and objectives we presented in our previous response - as well as the outline of our governance model and the additional kernels - already provide some useful information to address your remaining concerns.
> > > >
> > > > &nbsp;
> > > >
> > > > Sincerely,
> > > >
> > > > The Authors

---

> > > > > ### Comment · Reviewer_oVt5 · 2023-08-21
> > > > > **Update after SAC comment https://openreview.net/forum?id=vzrA6uqOis&noteId=1wstuXSBjW**
> > > > >
> > > > > I am very happy to see that GAUCHE has been used by a variety of researchers. This removes the possible suspicion (mine and potential readers) that this is a paper about a personal toolbox. Please do include these examples as part of the introduction of your paper!
> > > > >
> > > > > I have increased my score from 4 to 7 accordingly. I will not go above because I don't think this library makes anything possible that was not before but rather simplifies the inferential workflow.

---

> > > > > > ### Author Response · Authors · 2023-08-22
> > > > > > **Many Thanks for the Response and Constructive Feedback during the Review Process**
> > > > > >
> > > > > > &nbsp;
> > > > > >
> > > > > > We would like to close out the discussion period by thanking the reviewer once again for all their work in responding and offering constructive feedback during the review process!
> > > > > >
> > > > > > &nbsp;
> > > > > >
> > > > > > Sincerely,
> > > > > >
> > > > > > The Authors

---

### Official Review · Reviewer_jdZg · 2023-07-04

**Soundness:** 2 fair
**Presentation:** 2 fair
**Contribution:** 2 fair
**Rating:** 3
**Confidence:** 4

**Summary:**

This paper presents a library for Gaussian processes on chemistry data. In this library, a number of kernels are implemented over chemical representations such as graphs, strings and bit vectors. Regression and Bayesian optimization experiments are shown using the library.


**Strengths:**

- GP has been widely used as a ML tool for Chemistry.
- A reliable library can lower the barrier for chemistry experts on using ML tools.


**Weaknesses:**

- All of the implemented kernels are known in the literature.
- Neurips is probably not the right venue for publishing software libraries.


**Questions:**

Neurips is probably not the right venue for publishing software libraries.

**Limitations:**

The limitation of the proposed method has been explicitly discussed.

---

> ### Author Rebuttal · Authors · 2023-08-10
>
> &nbsp;
>
> Thank you for taking the time to review our manuscript. We were happy to see you emphasize both the general usefulness of Gaussian Process models as robust molecular machine learning tools, as well as the practical impact that a well-designed and easy-to-use library can have by making them more accessible to scientific experts in chemistry and materials science.
>
> The concerns you raised in your review relate to the suitability of NeurIPS as a venue for software library papers and the originality of the kernels we have implemented. We will address each of these points in turn below.
>
> &nbsp;
>
> ## __Publishing Software Library Papers at NeurIPS__ ##
>
> &nbsp;
>
> > Neurips is probably not the right venue for publishing software libraries.
>
> &nbsp;
>
> Our interpretation of the NeurIPS 2023 Call For Papers, which explicitly calls for “libraries” in the Infrastructure section, as well as a call for manuscripts on “Machine learning for sciences”, leads us to believe that our contribution falls within the remit of the current iteration of NeurIPS, albeit we acknowledge that the aforementioned calls are a new addition in recent years. We refer to [1-3] as examples of software libraries that were recently published at NeurIPS.
>
> &nbsp;
>
> ## __Originality of our Kernels__ ##
>
> &nbsp;
>
> > All of the implemented kernels are known in the literature.
>
> &nbsp;
>
> Building on the previous point, we would like to emphasize that our main contribution is the provision of a robust and easy-to-use library to make Gaussian Processes more easily accessible to expert chemists and materials scientists. While we strongly believe that the development of novel and more performant kernels is an important area of future research, we would like to point out that this is an orthogonal gap in the literature to the one we aim to address with this manuscript - though one that could strongly benefit from the robust foundation we aim to provide with GAUCHE.
>
> &nbsp;
>
> We hope that these additional clarifications address all of your questions and concerns. Please let us know if you have any further questions!
>
> &nbsp;
>
> Sincerely,
>
> The Authors
>
> &nbsp;
>
> ## __References__ ##
>
> &nbsp;
>
> [1] Jamasb, Arian, et al. “Graphein-a python library for geometric deep learning and network analysis on biomolecular structures and interaction networks.” Advances in Neural Information Processing Systems 35 (2022): 27153-27167.
>
> [2] Pineda, Luis, et al. “Theseus: A library for differentiable nonlinear optimization.” Advances in Neural Information Processing Systems 35 (2022): 3801-3818.
>
> [3] Feydy, Jean, et al. “Fast geometric learning with symbolic matrices.” Advances in Neural Information Processing Systems 33 (2020): 14448-14462.
>
> &nbsp;

---

### Official Review · Reviewer_rK4e · 2023-07-05

**Soundness:** 4 excellent
**Presentation:** 4 excellent
**Contribution:** 4 excellent
**Rating:** 8
**Confidence:** 4

**Summary:**

The authors present a framework called GAUCHE with comprehensive exploration of Gaussian Processes (GP) and their application to molecular machine learning. The authors thoroughly examine different ways of representing molecular structures - through hand tuned fingerprints, string notations (SMILES/SELFIES/Protein sequences), and undirected graphs. The authors evaluate the proposed kernels and representation schemes across several diverse benchmarks involving regression tasks, uncertainty quantification, and Bayesian optimization (BO). The experiments use diverse datasets, representing a span of property prediction from single molecule and reaction yield prediction from sets of molecules. Finally, the authors evaluate the best-performing GAUCHE kernels in a Bayesian optimization framework using three distinct datasets. Results reveal that GAUCHE's kernels outperforms random baseline, especially in low data regime, highlighting its practical utility in aiding chemists to prioritize synthesis candidates. Overall, this is an impressive work paper with strong theoretical framework with rigorous empirical evaluation.

**Strengths:**

1. This paper introduces a novel, theoretically robust kernel design for Gaussian process regression tailored for molecular data types.
2. The authors demonstrated the utility of the GAUCHE kernels in Bayesian optimization tasks which highlights their real-world applicability, particularly in the context of supporting chemists in candidate selection for synthesis.
3. The authors ensure reproducibility by providing clear explanations and thorough empirical evaluations as well as a very well designed codebase.
4. The paper uses a variety of datasets ranging from properties applicable to drug discovery and material discovery domain, which  showcases the adaptability of the GAUCHE kernels across different data.
5. The authors evaluate the GAUCHE kernels across a wide array of molecular input types ranging from fingerprints, strings and graphs highlighting the versatility of the proposed framework.

**Weaknesses:**

This paper presents a strong methodological framework and robust empirical evaluation, but it does fall short in some key areas.

1. The authors have shown that GAUCHE framework has been proven to be effective in a number of experiments, the comparison against SOTA neural network is notably lacking. For instance, chemprop, and ChemBERTa.
2. The authors emphasizes GAUCHE's applicability for a low data regime, which is certainly a critical need in many scientific domains. However, in cases where large, high-quality datasets are available. For instance, internal databases at big pharma companies or public repositories like BindingDB. It would be great to see experiments around how the computational requirements, training time, and model performance would be impacted when scaling up the dataset size.
3. It would be great if the authors could explore comparisons between RDKit/FragPrints and proprietary fingerprint techniques such as Dragon FP and Schrödinger, provided the licenses and costs are manageable.

**Questions:**

1. How does the splitting strategy affect the results? For example, what if the data were split based on molecular scaffolds, which would ensure the structural diversity of the test set, instead of a random split?
2. In the context of the Uncertainty Quantification benchmark, how does the predicted variance of the model relate to OOD samples? Could you provide a plot of the Tanimoto distance to the training dataset versus the predicted variance?
3. The authors touch upon the potential use of embeddings of molecules from pretrained models as inputs for the kernels, but there doesn't seem to be any results for this approach. Is this something the authors are considering for future investigations?
4. How would the performance of the kernels be impacted with the changes in the hand tuned fingerprint radius? Have any ablation studies been conducted to investigate this?

**Limitations:**

1. The GPR models can be computationally intensive, especially for larger datasets if used in conjunction with large dimensional fingerprints etc. This could limit the applicability in scenarios where abundant high-quality data is available.
2. The performance of predictive models in low data regimes is heavily affected by the choice of dataset splitting method. The results obtained using random splitting might not hold if other splitting methods like scaffold or temporal splitting are used.

---

> ### Author Rebuttal · Authors · 2023-08-10
>
> &nbsp;
>
> Thank you for taking the time to review our manuscript and for providing detailed, helpful and constructive feedback. We were happy to see you emphasizing the quality of our codebase and empirical evaluation, as well as the practical importance of our work.
>
> The main suggestions you raised in your review relate to a characterization of the computational requirements as the dataset size increases, as well as a comparison to state-of-the-art deep learning algorithms and proprietary fingerprints. We sincerely appreciate the feedback and will try to address your points below.
>
> &nbsp;
>
> ## __Computational Requirements for Larger Datasets__ ##
>
> &nbsp;
>
> > The authors emphasizes GAUCHE’s applicability for a low data regime, which is certainly a critical need in many scientific domains. However, in cases where large, high-quality datasets are available. For instance, internal databases at big pharma companies or public repositories like BindingDB. It would be great to see experiments around how the computational requirements, training time, and model performance would be impacted when scaling up the dataset size.
>
> > The GPR models can be computationally intensive, especially for larger datasets if used in conjunction with large dimensional fingerprints etc. This could limit the applicability in scenarios where abundant high-quality data is available.
>
> &nbsp;
>
> While black-box optimization tasks in low-data regimes are of substantial practical importance, we agree that it would be very interesting to investigate and extend GAUCHE to work well on larger datasets - for example, as you suggest, large public or private databases or even large DFT- or MD-derived datasets. As standard Gaussian Process inference scales cubically in the number of datapoints, it is likely too expensive to be a viable option in such settings. We have added a tutorial featuring the use of sparse GPs on the lipophilicity dataset (ca. 4200 molecules), which reduce this complexity by only performing inference over a subset of inducing points.
>
> &nbsp;
>
> ## __Comparison to Proprietary Fingerprints__ ##
>
> &nbsp;
>
> > It would be great if the authors could explore comparisons between RDKit/FragPrints and proprietary fingerprint techniques such as Dragon FP and Schrödinger, provided the licenses and costs are manageable.
>
> While we have not explored the cost and licensing considerations of proprietary molecular fingerprints, we hope that the modularity of GAUCHE helps to facilitate such a comparison, as it would be straight-forward to apply any of our kernels to a dataset of molecules featurized as Dragon or Schrödinger fingerprints. At the reviewer's suggestion to expand the set of featurizations, we have now included Mordred descriptors which achieve state-of-the-art performance on the photoswitch benchmark. We include these results in the attached pdf.
>
> &nbsp;
>
> ## __Comparison to SOTA Neural Networks__ ##
>
> &nbsp;
>
> > The authors have shown that GAUCHE framework has been proven to be effective in a number of experiments, the comparison against SOTA neural network is notably lacking. For instance, chemprop, and ChemBERTa.
>
> &nbsp;
>
> As most SOTA deep learning frameworks (such as Chemprop and ChemBERTa) do not provide out-of-the-box support for BO/UQ it may be difficult to include them directly. However, the ChemBERTa embeddings could be added as an additional featurization. Additionally, we agree that it would be interesting to identify uncertainty-aware and BO-compatible deep learning frameworks for future benchmarking studies.
>
> &nbsp;
>
> ## __Questions__ ##
>
> &nbsp;
>
> > How does the splitting strategy affect the results? For example, what if the data were split based on molecular scaffolds, which would ensure the structural diversity of the test set, instead of a random split?
>
> > In the context of the Uncertainty Quantification benchmark, how does the predicted variance of the model relate to OOD samples? Could you provide a plot of the Tanimoto distance to the training dataset versus the predicted variance?
>
> &nbsp;
>
> This is an excellent point. While we have chosen random splits for our current experimental setup to mimic late-stage molecular optimization within a chemical series, we agree that characterizing and comparing the predictive accuracy and calibration of different models in an out-of-distribution regime is a useful additional consideration. We will try to set up the corresponding experiments and add them to the appendix of our manuscript. For the Tanimoto kernel, given that it is using the Tanimoto distance metric directly, we would expect there to be a direct correlation albeit it is unclear what we would expect for non-binary representations.
>
> &nbsp;
>
> > The authors touch upon the potential use of embeddings of molecules from pretrained models as inputs for the kernels, but there doesn’t seem to be any results for this approach. Is this something the authors are considering for future investigations?
>
> &nbsp;
>
> There is a tutorial notebook, *pretrained_kernel.py* in the notebooks folder of the library that considers the case of pretrained embeddings. Seeing as this methodology is of interest, we can run more extensive experiments and include them as an additional point of comparison.
>
> &nbsp;
>
> > How would the performance of the kernels be impacted with the changes in the hand tuned fingerprint radius? Have any ablation studies been conducted to investigate this?
>
> &nbsp;
>
> We have not carried out any kernel-specific ablations yet, but agree that this would be another interesting plot to add to our appendix, as it would illustrate how the performance of Gaussian Process models changes when they are trained on progressively more expressive molecular representations.
>
> &nbsp;
>
> ## __Summary__ ##
>
> &nbsp;
>
> We hope that this additional discussion addresses all of your points. Please let us know if you have any further questions!
>
> &nbsp;
>
> Sincerely,
> The Authors
>
> &nbsp;

---

> > ### Comment · Reviewer_rK4e · 2023-08-16
> >
> > I appreciate the authors' comprehensive response to the feedback and the steps taken to address the main points raised in the review. The rebuttal addresses my concerns thoroughly, especially in regards to computational requirements for larger datasets, comparison with SOTA methods and the comparison to proprietary fingerprints. However, specific details on why direct comparison with some neural networks might not be applicable could strengthen your argument. Also, the discussion on the computational requirements is comprehensive, and the rebuttal carefully outlined the approach to proprietary fingerprints.
> >
> > Regarding the questions, the authors have provided mostly satisfactory answers, with clear acknowledgments and plans for further investigation. My current scores for this paper remain the same at this stage.

---

> > > ### Author Response · Authors · 2023-08-18
> > > **Additional Experiments and Clarification**
> > >
> > > Thank you for your response! We are happy to hear that our rebuttal thoroughly addressed your questions and concerns. In the following, we would like to clarify our response regarding the direct comparison to deep neural networks and share results from three additional experiments motivated by your suggestions.
> > >
> > > &nbsp;
> > >
> > > # __Direct Comparison to Deep Learning Algorithms__
> > >
> > > &nbsp;
> > >
> > > We would like to refer to Appendix A to point out that we do already perform an extensive empirical comparison to a range of state-of-the-art uncertainty-aware deep neural networks [1-4]. We apologise for any confusion and will make sure to feature these results more prominently in the main text of our manuscript, as we agree that they provide an important reference point.
> > >
> > > In our previous response, we were referring to the fact that the predictions of deep learning frameworks such as ChemProp and ChemBERTa do not provide native estimates of their predictive uncertainty. However, we discovered that ChemProp does provide an ensembling functionality that produces empirical uncertainty estimates that we can directly compare against our GPs.
> > >
> > > We have thus carried out an additional empirical evaluation of ChemProp on the Photoswitch dataset. Specifically, we created 20 random train-test splits, for each of which we performed 100 iterations of `hyperopt`-based hyperparameter search using the functionalities provided in ChemProp. Using the best hyperparameter combination, we then trained an ensemble of five GNNs and evaluated it on each held-out test set, using the mean and variance of the ensemble predictions to compute the root-mean-squared error (RMSE) and negative log-predictive density (NLPD) to quantify the models' predictive accuracy and the calibration of their predictive uncertainty estimates. As is apparent from the table below, we found the ensembling approach of ChemProp to underperform a Tanimoto-kernel GP.
> > > | Method | RMSE (&darr;)  | NLPD (&darr;)  |
> > > | -------- | -------- | -------- |
> > > | ChemProp|30.35&pm;1.30|4.53&pm;0.83|
> > > | Tanimoto GP |**20.9&pm;0.7**|**0.22&pm;0.03**|
> > >
> > > &nbsp;
> > > # __Results for Scaffold Splits__
> > > &nbsp;
> > > > How does the splitting strategy affect the results? For example, what if the data were split based on molecular scaffolds, which would ensure the structural diversity of the test set, instead of a random split?
> > >
> > > To answer this question, we have re-run parts of our experimental evaluation with 80-20 Bemis-Murcko scaffold splits instead of random splits. As only the lipophilicity dataset exhibits sufficient scaffold diversity to perform such an analysis (the skewness of the scaffold distribution in the others makes an 80-20 split impossible), the following results focus on the predictive accuracy (RMSE) and calibration (NLPD) of GP models in this setting.
> > >
> > > While this more challenging evaluation setup leads to slightly higher RMSEs and NLPDs, we note that one can observe the same trends as with random splits: Tanimoto-based GPs generally outperform Scalar Product ones, while string kernel-based GPs are better than both.
> > >
> > > |Kernel| Representation | RMSE (&darr;)| NLPD (&darr;)|
> > > |:---:|:---:|:---:|:---:|
> > > |Tanimoto|Fragprints| 0.86 &pm; 0.01 | **1.02 &pm; 0.04** |
> > > ||Fingerprints| 0.88 &pm; 0.01 | 1.12 &pm; 0.04 |
> > > ||Fragments| 0.89 &pm; 0.01 | 2.10 &pm; 0.13 |
> > > | Scalar Product |   Fragprints   | 0.89 &pm; 0.01 | 1.75 &pm; 0.08 |
> > > ||Fingerprints| 0.95 &pm; 0.01 | 1.99 &pm; 0.09 |
> > > ||Fragments| 1.00 &pm; 0.01 |      NaN       |
> > > |String|SMILES| **0.82 +- 0.01**  |  1.08 +- 0.04  |
> > >
> > > &nbsp;
> > > # __Ablation over Fingerprint Radius__
> > > &nbsp;
> > >
> > > > How would the performance of the kernels be impacted with the changes in the hand tuned fingerprint radius? Have any ablation studies been conducted to investigate this?
> > >
> > > Motivated by your suggestion, we have performed an ablation study over the hand-tuned radius parameter of extended-connectivity fingerprints (ECFPs). Specifically, we trained Tanimoto-kernel GPs on the Photoswitch dataset using a series of five increasing radius parameters. In the table below, we report the mean and standard error of the RMSE and NLPD over 50 different 80-20 train-test-splits.
> > >
> > > Intriguingly, these results show a strong negative correlation between the fingerprint radius and predictive performance. We hypothesize that this is caused by the fact that an expanding feature space leads to lower and less informative Tanimoto similarity scores, making it more difficult to train generalisable models.
> > >
> > > | Fingerprint |RMSE (&darr;)|NLPD (&darr;)|
> > > |:---:|:---:|:---:|
> > > |ECFP4|**22.65&pm;0.55**|**0.41&pm;0.05**|
> > > |ECFP6|23.50&pm;0.55|0.47&pm;0.05|
> > > |ECFP8|24.43&pm;0.54|0.52&pm;0.05|
> > > |ECFP10|25.17&pm;0.54|0.56&pm;0.04|
> > > |ECFP12|25.70&pm;0.53|0.58&pm;0.04|
> > >
> > > &nbsp;
> > >
> > > We thank you again for the very helpful feedback and hope that these additional clarifications and experimental results are helpful in addressing any remaining concerns. Please let us know if you have any further questions.
> > >
> > > Sincerely,
> > >
> > > The Authors

---

> > > > ### Comment · Reviewer_rK4e · 2023-08-20
> > > >
> > > > Thanks authors for an another great response. I am impressed by your ability to conduct the necessary experiments so rapidly and respond comprehensively to the points raised in my previous response. In light of these ablation studies, I am updating my Soundness rating for this work to a 4.
> > > >
> > > > Thank you for the thorough and responsive engagement with the review process.

---

> > > > > ### Author Response · Authors · 2023-08-21
> > > > > **Thank You for the Extensive and Constructive Feedback during the Review Process**
> > > > >
> > > > > &nbsp;
> > > > >
> > > > > We wish to take this opportunity to thank the reviewer for their extensive constructive feedback during the review process, for raising interesting and relevant questions for the work, and for highlighting a number of action points which have served to strengthen the paper and will benefit users of the GAUCHE library!
> > > > >
> > > > > &nbsp;
> > > > >
> > > > > Sincerely,
> > > > >
> > > > > The Authors

---

### Author Rebuttal · Authors · 2023-08-10

&nbsp;

## __Overview__ ##

&nbsp;

We would like to thank all reviewers for the time and effort put into reviewing our manuscript and for the valuable and constructive feedback they have provided.

We are delighted that all reviewers recognized the practical significance of our work, highlighting that GAUCHE “fills a gap in the open-source GP stack” (**reviewer oVt5**) and provides a “strong theoretical framework with rigorous empirical evaluation” (**reviewer rK4e**) that “allows non-GP-expert chemists to utilize state-of-the-art black-box optimization tools” (**reviewers JWPZ and jdZg**) and is able to “adapt to proteins, molecules, and chemical reactions” (**reviewer rHxF**).

We are also happy to see that reviewers appreciated our scope, noting that we “provide thorough explanations of molecular and chemical reaction” tasks (**reviewer JWPZ**), “thoroughly examine different ways of representing molecular structures” (**reviewer rK4e**) and “properly analyze the related kernel functions” (**reviewer rHxF**), which we evaluate “across several diverse benchmarks involving regression tasks, uncertainty quantification, and Bayesian optimization (BO)” (**reviewer rK4e**).

Finally, we were pleased to hear that reviewers appreciated our engineering effort, noting that we provide “a very well designed codebase” (**reviewer rK4e**) that “seems easy to follow and adapt” (**reviewer rHxF**), further emphasizing that “the code cleanliness is rather high and the code is well-tested” (**reviewer oVt5**) and that we “ensure reproducibility by providing clear explanations” and tutorials (**reviewer rK4e**).

&nbsp;

## __Summary of Concerns__ ##

&nbsp;

The main concerns raised by reviewers related to:

&nbsp;

1. The suitability of NeurIPS as a venue for publishing software libraries (**reviewer jdZg**),
2. The need for a clearer emphasis of the library’s strengths (**reviewers JWPZ and rHxF**) and
3. Examples of real-world case studies that use GAUCHE (**reviewer oVt5**).

&nbsp;

**(1)** We have addressed the first point by referring to the NeurIPS Call For Papers - which explicitly invites contributions on “libraries” (and “machine learning for sciences”) - as well as a range of software libraries that were recently published at NeurIPS [1-3].

**(2)** In response to the feedback we received regarding the second point, we will further refine our abstract, introduction and discussion of prior art to more clearly highlight the strengths and added value that GAUCHE provides (namely ease-of-use, modularity and robust empirical performance). As requested by reviewers, we have also summarized the advantages our library provides over existing packages in the table below, which we will add to Section 5.

&nbsp;

| Library  | Gaussian Processes | Bayesian Optimisation | Molecular Representations | Chemistry Tutorials | Graph Kernels | Bit Vector Kernels | String Kernels |
|----------|--------------------|-----------------------|---------------------------|---------------------|---------------|--------------------|----------------|
| GPyTorch | ✓                  | ✗                     | ✗                         | ✗                   | ✗             | ✗                  | ✗              |
| GPflow   | ✓                  | ✗                     | ✗                         | ✗                   | ✗             | ✗                  | ✗              |
| BoTorch  | ✓                  | ✓                     | ✗                         | ✗                   | ✗             | ✗                  | ✗              |
| DeepChem | ✗                  | ✗                     | ✓                         | ✓                   | ✗             | ✗                  | ✗              |
| GraKel   | ✗                  | ✗                     | ✗                         | ✗                   | ✓             | ✗                  | ✗              |
| FlowMO   | ✓                  | ✗                     | ✓                         | ✓                   | ✗             | ✓                  | ✓              |
| GAUCHE   | ✓                  | ✓                     | ✓                         | ✓                   | ✓             | ✓                  | ✓              |

&nbsp;

**(3)** In response to the request for real-world case studies, we are eager to share at least four instances (for which public references are available) in which GAUCHE has been used by other researchers in real-world research and production settings. As the review guidelines state that we should not include any links to external pages, we will liaise with the AC/SAC to determine the best way to provide these references.

We have added additional discussions and clarifications of these and all other points raised by the reviewers under the respective reviews. Additionally, our attached rebuttal document includes a more polished version of the markdown table above as well as new results attaining a new **SOTA on the photoswitch benchmark** following the suggestions of **reviewer rK4e**.

We hope that our response addresses all reviewer questions and concerns and are happy to answer any further questions!

&nbsp;

Sincerely,

The Authors

&nbsp;

## __References__ ##

&nbsp;

[1] Jamasb, Arian, et al. “Graphein-a python library for geometric deep learning and network analysis on biomolecular structures and interaction networks.” Advances in Neural Information Processing Systems 35 (2022): 27153-27167.

[2] Pineda, Luis, et al. “Theseus: A library for differentiable nonlinear optimization.” Advances in Neural Information Processing Systems 35 (2022): 3801-3818.

[3] Feydy, Jean, et al. “Fast geometric learning with symbolic matrices.” Advances in Neural Information Processing Systems 33 (2020): 14448-14462.

&nbsp;

---

> ### Author Response · Authors · 2023-08-21
>
> As the author discussion period is coming to an end, we would like to thank all reviewers once again for the time and effort they have dedicated to reviewing our manuscript and engaging with our rebuttal. We were pleased to see that our rebuttals and subsequent discussion led to a significant increase in scores and greatly appreciate the detailed and helpful feedback that reviewers have provided.
>
> Sincerely,
>
> The Authors

---

### Decision · Program_Chairs · 2023-09-21

**Decision:**

Accept (poster)

**Comment:**

The authors provide a new library for GAUssian processes in CHEmistry.

The paper is well-written and easy to follow.

As indicated by their scores, the reviewers really liked the paper and the reviewers noted that GAUCHE “fills a gap in the open-source GP stack”, it “allows non-GP-expert chemists to utilize state-of-the-art black-box optimization tools”, and it is “very well designed codebase”.

The only negative concern from one of the reviewers was that NeurIPS might not be the best fit for this paper since it is not a traditional NeurIPS paper in the sense that it is rare to see software libraries published at NeurIPS.

The authors responded with a citation from the NeurIPS call from papers where it is listed that NeurIPS welcomes software libraries and they also listed some recent examples where NeurIPS published software libraries:

[1] Jamasb, Arian, et al. “Graphein-a python library for geometric deep learning and network analysis on biomolecular structures and interaction networks.” Advances in Neural Information Processing Systems 35 (2022): 27153-27167.

[2] Pineda, Luis, et al. “Theseus: A library for differentiable nonlinear optimization.” Advances in Neural Information Processing Systems 35 (2022): 3801-3818.

[3] Feydy, Jean, et al. “Fast geometric learning with symbolic matrices.” Advances in Neural Information Processing Systems 33 (2020): 14448-14462.

The GAUCHE software package indeed might be a great tool for working on certain chemistry problems with machine learning and the authors also provided a list of papers from various authors where this library has been already used.